# Cellular cartography of the organ of Corti based on optical tissue clearing and machine learning

**Shinji Urata[1,2], Tadatsune Iida[1], Masamichi Yamamoto[3], Yu Mizushima[2], Chisato Fujimoto[2], Yu Matsumoto[2], Tatsuya Yamasoba[2]\*, Shigeo Okabe[1]\***

[1]Department of Cellular Neurobiology, Graduate School of Medicine, The University of Tokyo, Tokyo, Japan; [2]Department of Otolaryngology, Graduate School of Medicine, The University of Tokyo, Tokyo, Japan; [3]Department of Nephrology, Graduate School of Medicine, Kyoto University, Kyoto, Japan

**Abstract** The highly organized spatial arrangement of sensory hair cells in the organ of Corti is essential for inner ear function. Here, we report a new analytical pipeline, based on optical clearing of tissue, for the construction of a single-cell resolution map of the organ of Corti. A sorbitol-based optical clearing method enabled imaging of the entire cochlea at subcellular resolution. High-fidelity detection and analysis of all hair cell positions along the entire longitudinal axis of the organ of Corti were performed automatically by machine learning–based pattern recognition. Application of this method to samples from young, adult, and noise-exposed mice extracted essential information regarding cellular pathology, including longitudinal and radial spatial characteristics of cell loss, implying that multiple mechanisms underlie clustered cell loss. Our method of cellular mapping is effective for system-level phenotyping of the organ of Corti under both physiological and pathological conditions.
DOI: https://doi.org/10.7554/eLife.40946.001

**\*For correspondence:**
tyamasoba-tky@umin.ac.jp (TY);
okabe@m.u-tokyo.ac.jp (SO)

**Competing interests:** The authors declare that no competing interests exist.

## Introduction

A complete understanding of auditory perception and transduction relies on an accurate reconstruction of the intact, three-dimensional structure of the cochlea. The spatial organization of the organ of Corti, the mammalian auditory sensory epithelium, determines the cochlear tonotopic map, which associates the positions of the inner hair cells (IHCs) in the cochlea with local characteristic frequencies. The basic pattern of the tonotopic map is simple, with higher frequencies on the base of the cochlear spiral and lower frequencies on the apex (*von Békésy and Peake, 1990*). However, multiple structural and cell biological factors influence the actual shape of the tonotopic map (*Temchin and Ruggero, 2014*).

In humans, age-related (*Chien and Lin, 2012*) and noise-induced hearing loss (*Nelson et al., 2005*) are prevalent health problems that require early prevention (*Cunningham and Tucci, 2017*). However, the field awaits the development of appropriate model animals that recapitulate human pathology (*Wang et al., 2002*; *Yamasoba et al., 1998*; *Zheng et al., 1999*). Moreover, the complexities of cochlear structure have prohibited a comprehensive cellular cartography, and current histological techniques are far from satisfactory for comprehensive analyses. Therefore, new methods that enable simultaneous examination of molecular signatures and subcellular structures across the entire cochlea would greatly accelerate the progress of research on auditory mechanisms.

Optical access to the properties of cells within highly complex tissues and organs is an important technical goal of modern cell biology. Advancements in optical tissue clearing have enabled the acquisition of structural and molecular information from large volumes of tissues and organs

(*Chung et al., 2013*; *Dodt et al., 2007*; *Hama et al., 2015*; *Renier et al., 2014*; *Susaki et al., 2014*). Recent reports showed that both organic solvent– and hydrophilic solution–based clearing methods could be optimized in clearing hard tissues that contain large proportions of extracellular matrix (*Berke et al., 2016*; *Cai et al., 2018*; *Calve et al., 2015*; *Greenbaum et al., 2017*; *Jing et al., 2018*; *Tainaka et al., 2018*; *Treweek et al., 2015*). The accumulating knowledge and technologies should be helpful in development of effective clearing and labeling protocols for the inner ear inside the temporal bone (*Nolte et al., 2017*; *Tinne et al., 2017*). To date, however, an integrated method of tissue processing, labeling, and imaging techniques with single cell resolution has not yet been developed and optimized for the inner ear.

Here, we report an analytical pipeline for the construction of a single-cell resolution map of the organ of Corti taken from C57BL/6J mice at postnatal day (PND) 5, 60, 120 and 360. The method is based on optical tissue clearing technology and automatic cell detection using a machine learning algorithm. In this method, a series of fixation, permeabilization, immunolabeling, and clearing processes transform the inner ear into optically transparent samples suitable for volume imaging at single-cell resolution. Automated high-fidelity recording and analysis of hair cell positions along the entire length of the organ of Corti were achieved based on machine learning–based cell detection. Application of this method to pathological samples revealed distinct impacts of aging and noise on spatial features of sensory hair cell pathology. Our method of cellular mapping is highly effective for system-level phenotyping of the organ of Corti.

## Results and discussion

### Analytical pipeline for cellular cartography of the organ of Corti

Our analytical pipeline for sensory hair cell mapping in the cochlea followed three steps. First, the mouse temporal bone was isolated at PND 60 and processed for clearing and immunolabeling of the sensory hair cells (*Figure 1A* and *Figure 1—figure supplement 1*). Tissue clearing and immunolabeling were optimized for tissue transparency and antibody accessibility. After tissue preparation, three-dimensional two-photon excitation microscopy generated image stacks covering the entire structure of the organ of Corti, with an average size of 1200 × 1200 × 800 μm (*Figure 1B*). The image data were processed by custom-made software to stitch and linearize the sensory epithelium, followed by detection of cell positions (*Figure 1C*). The software automatically generated a spatial map of the total hair cells and estimated the positions of putative lost cells. The entire experimental procedure could be completed within 5 days, with 4 days for tissue clearing and labeling, 4 hr for image acquisition, and 30 min for automated analysis.

### Optimization of imaging of the whole intact cochlea

The mouse inner ear forms complex and intricate structures inside the temporal bone. To achieve deeper and clearer imaging of the organ of Corti, it was necessary to overcome two hurdles. First, an inorganic component of the bone, mainly composed of calcium phosphate, had to be removed. Second, refractive index (RI) matching had to be fine-tuned to decrease optical aberration induced by heterogenous tissue components (*Acar et al., 2015*; *Berke et al., 2016*). The existence of multiple methods for optical tissue clearing provided us with an opportunity to perform a side-by-side comparison of their applicability to the organ of Corti. We tested five independent, well-established clearing and labeling protocols (3DISCO (*Ertürk et al., 2012*), iDISCO (*Renier et al., 2014*), CLARITY (*Chung et al., 2013*), CUBIC (*Susaki et al., 2014*), and Sca*l*eS (*Hama et al., 2015*)) for their performance in detection of total hair cells. Myosin 7a (MYO7A), specifically expressed in IHCs and outer hair cells (OHCs), was utilized as a standard marker for hair cells. We found that performances of different protocols were comparable when they were applied to adult mouse brains, but the efficiencies for clearing the temporal bone were variable (*Figure 1D* and *Figure 1—figure supplement 1*). We failed to detect MYO7A-immunopositive hair cells in samples processed by 3DISCO, iDISCO, CLARITY, or CUBIC (*Figure 1D*). Microdissection of the membrane labyrinth of the iDISCO-processed samples confirmed the presence of MYO7A-immunopositive hair cells, suggesting that the surrounding bone tissue prevented the detection of fluorescence (*Figure 1E*). When CUBIC was combined with decalcification, MYO7A fluorescence could be detected down to 180 μm from the surface, but the combined method still did not enable imaging of the deeper part of the organ of

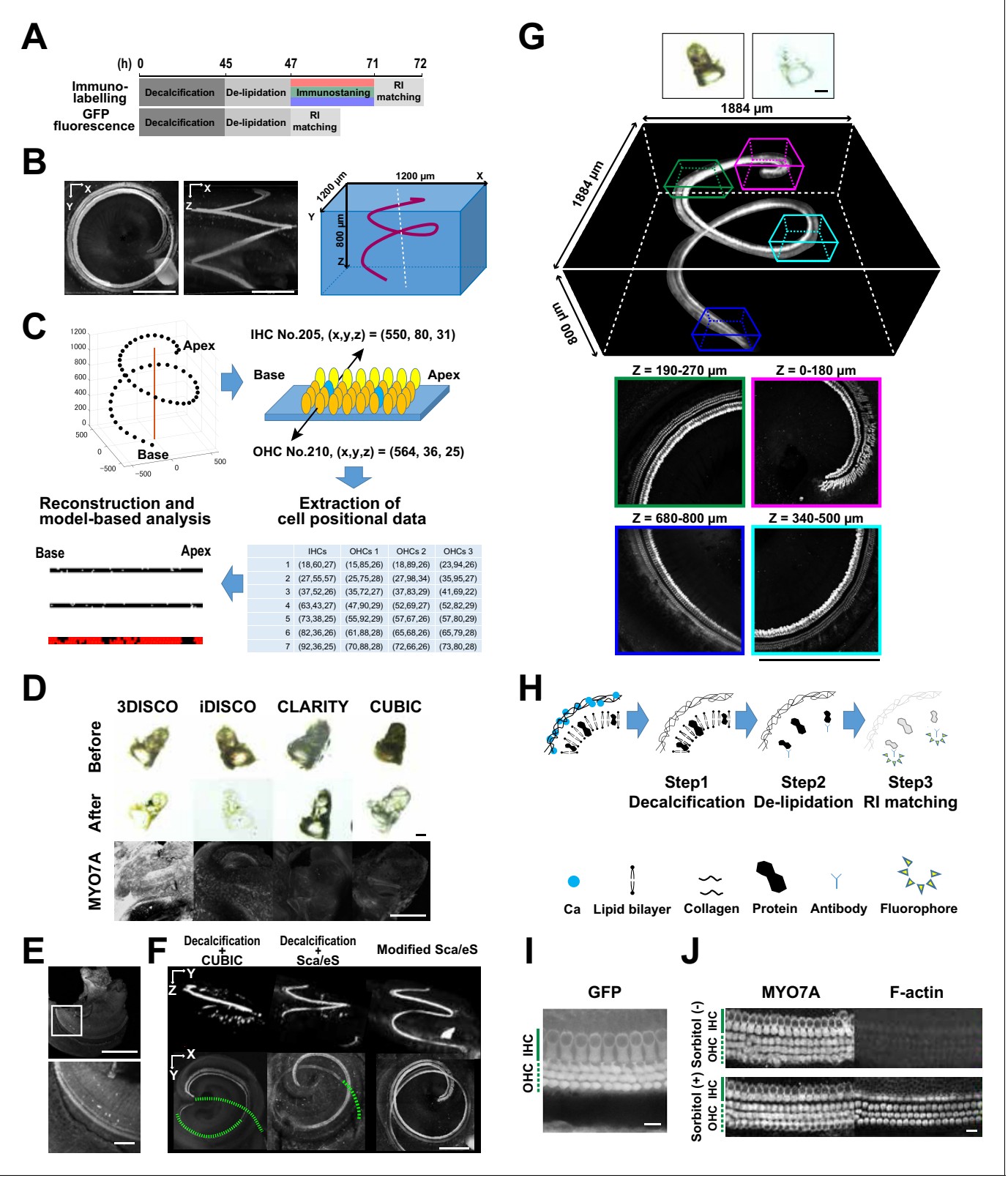

**Figure 1.** Optical tissue clearing and whole-mount immunolabeling of the organ of Corti. (**A**) Time course and individual steps of tissue clearing with or without immunostaining. (**B**) Three-dimensional imaging of the organ of Corti within the temporal bone. Top view (left), lateral view (middle), and the schematic presentation of the organ of Corti with its axis parallel to the modiolus. The size of the organ of Corti is indicated in the X, Y, and Z coordinates. Scale bar, 500 μm. (**C**) Computational processes of linearization, cell detection, and modeling. (**D**) Side-by-side comparison of 3DISCO,

*Figure 1 continued on next page*

*Figure 1 continued*

iDISCO, CLARITY, and CUBIC. Transmitted light images of samples before and after clearing, together with MYO7A staining. Scale bar, 500 µm. (E) Manual dissection of the iDISCO-processed sample confirmed MYO7A staining in the sensory epithelium. Scale bars, 500 µm (upper image) and 100 µm (lower image). (F) Lateral and horizontal views of the reconstructed three-dimensional images of the organ of Corti stained with anti-MYO7A. CUBIC with decalcification and original Sca*l*eS failed to detect the deepest part of the organ of Corti (green dotted lines). With modified Sca*l*eS, the entire structure of the organ of Corti could be visualized. Scale bar, 500 µm. (G) Modified Sca*l*eS sample of the organ of Corti stained with anti-MYO7A antibody, together with transmitted light images before (upper left) and after (upper right) treatment. Scale bar, 500 µm. (H) Three steps of the modified Sca*l*eS protocol. The initial decalcification step is followed by a clearing step, which mainly removes lipids from the extracellular matrix. Finally, the RI of the sample is matched with mounting solution. (I) Preservation of GFP fluorescence after modified Sca*l*eS treatment. Scale bar, 10 µm. (J) Preservation of rhodamine-phalloidin signal after modified Sca*l*eS treatment, which includes sorbitol to stabilize cytoskeletal polymers. Scale bar, 10 µm. IHC, inner hair cell; OHC, outer hair cell; RI, refractive index.

DOI: https://doi.org/10.7554/eLife.40946.002

The following source data and figure supplements are available for figure 1:

**Source data 1.** Source data for *Figure 1B, E* and *Figure 1—figure supplement 2*.

DOI: https://doi.org/10.7554/eLife.40946.005

**Figure supplement 1.** Protocol of modified Sca/eS.

DOI: https://doi.org/10.7554/eLife.40946.003

**Figure supplement 2.** Application of modified Sca*l*eS to other tissues.

DOI: https://doi.org/10.7554/eLife.40946.004

Corti (*Figure 1F*). Among the pre-existing tissue clearing methods, Sca*l*eS combined with decalcification yielded the best results (*Figure 1F*). However, this method still missed hair cells at the cochlear base, more than 500 µm away from the bone surface.

By modifying the original Sca*l*eS method, we achieved efficient in situ detection of all MYO7A-positive hair cells in the organ of Corti (*Figure 1G* and *Figure 1—figure supplement 1*). In the new protocol, we first decalcified the samples with EDTA (*Figure 1H*). In the subsequent clearing step, a combination of a nonionic detergent (Triton X-100) and an ionic chaotropic reagent (guanidine) was effective in increasing transparency. The Sca*l*eS and CUBIC1 protocols use urea instead of guanidine (*Hama et al., 2015*; *Susaki et al., 2015*), but high concentrations of urea can induce tissue expansion (*Tainaka et al., 2016*). By contrast, our guanidine-based clearing solution did not induce detectable tissue expansion (*Figure 1—figure supplement 1*). Although guanidine treatment denatures GFP and reduces its fluorescence intensity (*Huang et al., 2007*), this fluorescence quenching could be reversed by incubation in phosphate-buffered saline (PBS). Finally, we tested the solutions with RIs from 1.41 to 1.56 for their performance in tissue clearing by measuring the maximal depth of detectable MYO7A-positive hair cells from the temporal bone surface (*Figure 1—figure supplement 1*). We found that a RI matching solution with a RI of 1.47 was most effective for detecting MYO7A-positive hair cells away from the bone surface. This RI lies between that of bone matrix (RI = 1.56) and tissue with scarce extracellular matrix (RI = 1.38). The new protocol for the in situ detection of all MYO7A-positive hair cells in the organ of Corti could be completed within 4 to 6 days (*Figure 1A,H*, and *Figure 1—figure supplement 1*), and effectively detected GFP-based reporter molecules and F-actin by rhodamine phalloidin (*Figure 1I,J*). The presence of sorbitol in the clearing solution improved F-actin stabilization. The protocol could also be applied to detection of cellular components in other types of bone-containing samples (*Figure 1—figure supplement 2*).

## Machine learning–based automated detection of sensory hair cells

To obtain information about hair cell distribution along the entire longitudinal axis of the organ of Corti, we applied our optimized tissue clearing and labeling methods to samples from naïve C57BL/6J mice, and detected sensory hair cells with anti-MYO7A antibody. The combination of a widely used marker of sensory hair cells (MYO7A) and a standard mouse line (C57BL/6J) should facilitate replication of this protocol in other laboratories and comparative studies. Multiple image stacks that cover the entire structure of the organ of Corti were obtained by two-photon microscopy with voxel sizes of $0.99 \times 0.99 \times 1.0$ µm for high-resolution imaging (*Figure 2A*). Spatially confined two-photon excitation effectively decreased photobleaching after repetitive imaging. To adjust the local fluorescence intensity of MYO7A-immunopositive hair cells, we controlled both excitation laser power and the cut-off range of pixel intensities.

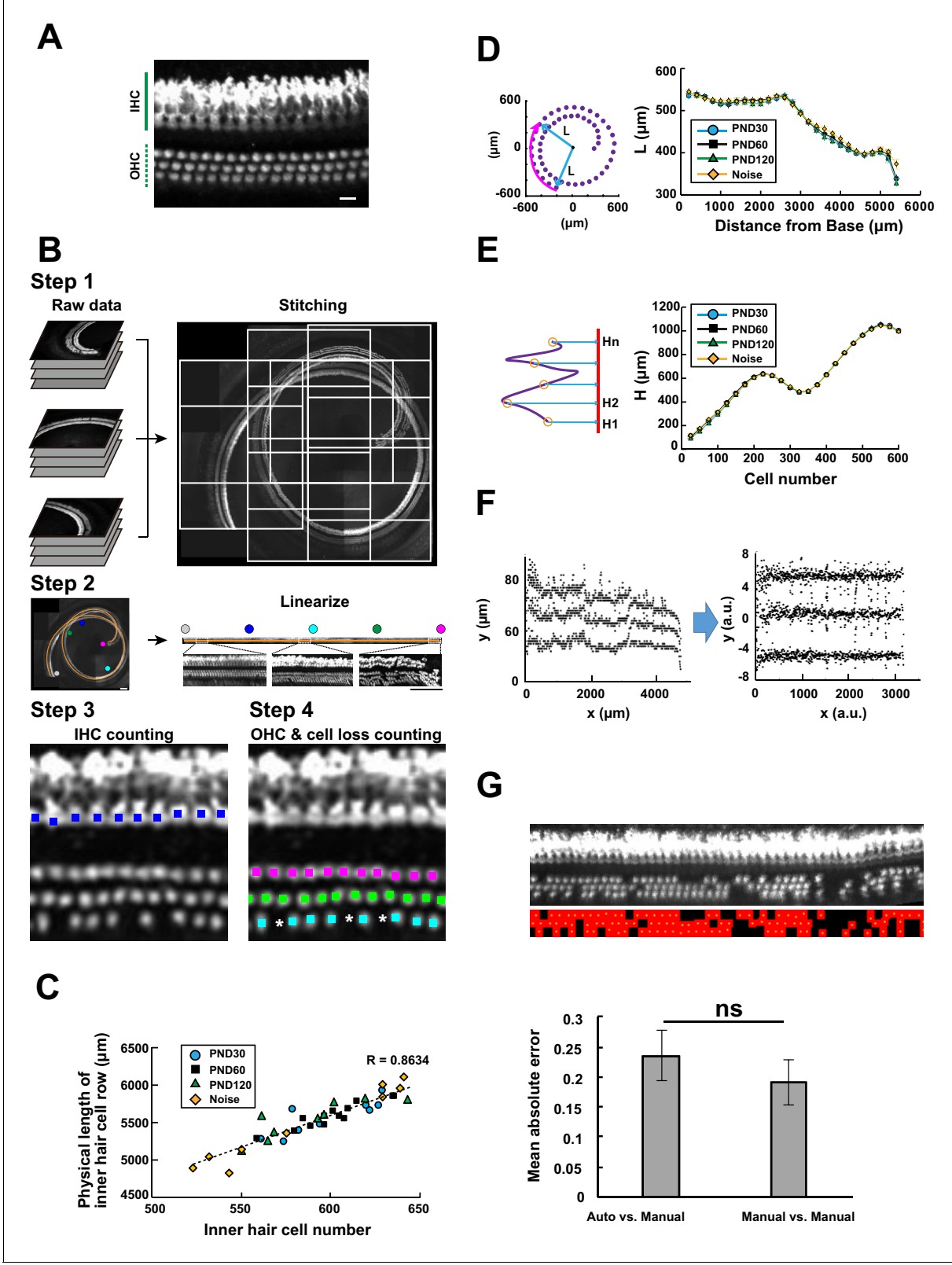

**Figure 2.** Computational analysis of hair cell distribution in the organ of Corti. (**A**) Detection of single hair cells stained with anti-MYO7A. The border between hair cells can be clearly detected. Scale bar, 10 μm. (**B**) Sequential steps in reconstruction of the linearized voxel image of the organ of Corti. The linearized voxel image was generated using the row of IHCs as a structural reference of the longitudinal axis of the organ of Corti. Scale bar, 100 μm. (**C**) Plot of the total longitudinal length of the organ of Corti against the total number of IHCs. (**D**) Plot of radial distance of IHCs from the modiolus. *Figure 2 continued on next page*

*Figure 2 continued*

(E) Plot of hair cell positions along the vertical axis of the organ of Corti. (F) Normalization of heterogeneity in hair cell positions. Before normalization, both x and y axes represent physical positions of OHCs. After normalization, the coordinates are arbitrary units and are equalized in x and y axes. (G) Transformation of the positions of hair cells to fit the standardized template. The template is a two-dimensional grid parallel to the surface of the sensory epithelium (upper image). This transformation is useful for estimation of lost hair cells based on the calculation of cell-free space. The accuracy of estimation by this method was comparable to the performance of manual estimation (lower plot). [n = 161 samples for each. Paired *t*-test; ns, not significant (p > 0.05).] IHC, inner hair cell; OHC, outer hair cell; PND, postnatal day; RI, refractive index.

DOI: https://doi.org/10.7554/eLife.40946.006

The following source data and figure supplements are available for figure 2:

**Source data 1.** Source data for *Figure 2D, E, G* and *Figure 2—figure supplement 1*.

DOI: https://doi.org/10.7554/eLife.40946.009

**Figure supplement 1.** Manual counting of lost hair cells and auditory brainstem-evoked response (ABR) in mice with age-related and noise-induced hearing loss.

DOI: https://doi.org/10.7554/eLife.40946.007

**Figure supplement 2.** Three-dimensional presentation of hair cell distribution projected to X-Y and Y-Z planes.

DOI: https://doi.org/10.7554/eLife.40946.008

To achieve automated detection of both IHCs and OHCs, we developed a series of custom-made MATLAB scripts (*Figure 2B* and *Table 1*, also see Appendices 1 and 2). Because loss of IHCs is rare even in pathological conditions, such as aging and noise exposure, the row of IHCs was used as a guide for linearization of the spiral sensory epithelium. First, multiple image stacks containing portions of the organ of Corti were assembled into a single image stack. Our image acquisition protocol was designed to obtain image stacks covering the volume of the entire cochlea. We also designed that the two adjacent imaged stacks always have the overlapping volume. With these image acquisition rules, the entire tissue volume containing the whole sensory epithelium could be easily reconstructed. Second, the best-fit arcs of the single IHC row were calculated to create a spiral that could be used as a structural reference for the entire organ of Corti. Third, a linearized voxel image was reconstructed using the best-fit spiral and the normal vectors of the plane fitted to the segments of the sensory epithelium. Finally, we employed machine learning models to perform an exhaustive search of all hair cells and recorded their positions as Cartesian coordinates. The search process by machine learning technique consists of two parts: the first step of signal-noise discrimination and the second step for the recovery of false negatives. The details are provided in *Table 1* and Appendix 2. (*LeCun et al., 1989*; *Friedman et al., 2000*; *Breiman, 2001*)

To test the ability of our automated cell detection protocol to reliably record hair cell positions, we studied its performance by comparing its outputs with manually identified OHC positions in four independent samples of the organ of Corti labeled with anti-MYO7A antibody. The automated detection protocol recovered 98.8 ± 0.6% of manually identified hair cells. In turn, 99.7 ± 0.2% of hair cells identified by the algorithm were also scored as hair cells by human operators, with the remaining 0.3% representing false positives (*Table 2*). The detection efficiency of our protocol was much higher than a standard imaging processing protocol based on three-dimensional watershed ((*Soille and Vincent, 1990*; *Table 2* and Appendix 2).

## Automated detection of hair cells in samples with hair cell pathology

Pathological changes in the sensory epithelium associated with aging or noise exposure can impair the hearing functions of the inner ear. Previous studies provided qualitative evidence showing that the cellular changes associated with age-associated or noise-induced hearing loss partially overlap, but also have distinct characteristics. C57BL/6J mice are widely used for aging research and exhibit the classic pattern of age-related hearing loss, with the loss of both hair cells and neurons starting from the base (*Hunter and Willott, 1987*). The increase in auditory brainstem–evoked response (ABR) threshold starts at the age of 10 weeks (*Ison et al., 2007*). Manual counting of lost cells in C57BL/6J mice at PND 5, 60, and 360 confirmed age-related cell loss (ACL) (*Figure 2—figure supplement 1*). For the assessment of noise-induced cell loss (NCL), we applied acoustic overexposure stimulus to C57BL/6J mice at PND 60 to induce a moderate threshold shift (*Figure 2—figure supplement 1*), as previously reported (*Mizushima et al., 2017*; *Tuerdi et al., 2017*). The cellular

**Table 1.** Details of machine learning models (related to **Figure 2**).

| Models | Type[‡] | Algorithm | Configuration | Use | Predictor |
|---|---|---|---|---|---|
| IHC[*] 1 | Binary | Gentle Boost | 300 classification trees | Reduction of noise | Area, barycentric coordinates, maximum correlation coefficients, maximum intensity, same data set of the nearest neighbor group and relative position of the nearest neighbor group |
| IHC[*] 2 | Binary | Random Forest | 300 classification trees | Detection of cells | Adding to the above, prediction score by 'IHC[*] 1' of itself and that of adjacent groups in six directions[¶], relative position of the adjacent groups, and cropped image[††] |
| OHC[†] 1 | Binary | Gentle Boost | 300 classification trees | Reduction of noise | Same as 'IHC[*] 1' |
| OHC[†] 2 | Binary | Random Forest | 300 classification trees | Detection of cells | Same as 'IHC[*] 2' |
| OHC[†] 3 | Multiclass | Convolutional Neural Network | From the input, convolutional layer (filter size 5, number 60), ReLU[§] layer, fully connected layer, Softmax Layer (three classes), and output. | Estimation of belonging row | Cropped image (39 × 69 pixels in width and height) |
| OHC[†] 4 | Binary | Convolutional Neural Network | From the input, convolutional layer (filter size 5, number 60), ReLU[§] layer, convolutional layer (filter size 5, number 20), ReLU[§] layer, fully connected layer, Softmax Layer (two classes), and output. | Detection of cells in spaces | Cropped image (39 × 69 pixels in width and height) |

*. IHC, inner hair cell.

†. OHC, outer hair cell.

‡. Classification type.

§. Rectified Linear Unit.

¶. Adjacent groups in direction of 0–60°, 60–120°, 120–180°, 180–240°, 240–300°, 300–360° with the y-axis as an initial line in the x-y plane.

††. Initial image size is 21 × 69 pixels in width and height. The image is resized in 7 × 23 then reshaped in 1 × 161.

DOI: https://doi.org/10.7554/eLife.40946.010

pathology varied from nearly normal appearance to severe hair cell damage in the basal end of the cochlea, and exhaustive screening of hair cell loss in this context should be useful.

To appropriately interpret cell position data from samples harvested under physiological and pathological conditions, it is necessary to evaluate variation in the morphology of the organ of Corti. Variation may also exist among samples collected under identical experimental conditions, potentially confounding data interpretation. To evaluate such variation in morphology, we performed the following three types of measurements (Appendix 2). First, we measured the total longitudinal length of the IHC row and the number of IHCs. The plot of IHC number vs. the length of the IHC row for multiple samples is useful for evaluating the longitudinal sizes of the organ of Corti (**Figure 2C**). The plot shows that data from young control samples (PND 30) scattered in the range of 550–650 IHCs, indicating the presence of physiological variation. Variation in the length and the cell number did not show a specific trend between samples from different ages. However, the data from noise-exposed mice exhibited a tendency of higher variation, potentially due to selective loss of hair cells at the basal end in some NCL samples. Second, we projected the position of IHCs onto a plane perpendicular to the modiolus and plotted the distance of IHCs from the modiolus

**Table 2.** Detection efficiency of hair cells (related to *Figure 2*).*

| | Inner hair cell | | | | |
|---|---|---|---|---|---|
| | Detect. (n)[†] | Undetect. (n)[‡] | Err. detect. (n)[§] | Recover Rate[¶] | Accuracy rate[††] |
| Our Method | 576 ± 33 | 13 ± 12 | 2 ± 2 | 0.979 ± 0.021 | 0.997 ± 0.003 |
| 3D Watershed | 424 ± 98 | 152 ± 82 | 110 ± 78 | 0.733 ± 0.149 | 0.818 ± 0.100 |

| | Outer hair cell | | | | |
|---|---|---|---|---|---|
| | Detect. (n)[†] | Undetect. (n)[‡] | Err. Detect. (n)[§] | Recover Rate[¶] | Accuracy rate[††] |
| Our Method[‡‡] | 1989 ± 133 | 24 ± 13 | 6 ± 4 | 0.988 ± 0.006 | 0.997 ± 0.002 |
| Principle 1 Only[§§] | 1925 ± 131 | 69 ± 41 | 16 ± 13 | 0.966 ± 0.021 | 0.992 ± 0.006 |
| 3D Watershed | 1493 ± 197 | 496 ± 111 | 760 ± 381 | 0.748 ± 0.064 | 0.682 ± 0.103 |

*. Data from 10 samples (PND30: two sample, PND60: three sample, ACL: two sample, NCL: three sample). Data are expressed as means ± SD.

†. Detection number.

‡. Undetected number.

§. Erroneous detection number.

¶. Recover rate of manually identified hair cells by the automated detection algorithm (almost synonymous with recall).

††. The number of hair cells identified by both manual and automated detection divided by the number of hair cells identified by automated detection (almost synonymous with precision).

‡‡.The proposed method in this study (principle 1 + principle 2).

§§. The method using the first half of the proposed method. For details please see 'Principles of auto-detection by machine learning' in Appendix 2.

DOI: https://doi.org/10.7554/eLife.40946.011

(*Figure 2D*). This plot revealed small variation in IHC position among experimental groups. Third, the positions of IHCs were projected onto the axis of the modiolus, and the relative positions of every 25 IHCs were plotted (*Figure 2E*). The IHC distributions along the axis of the modiolus in all samples were similar. In summary, these measurements confirmed that the overall spatial distribution of IHCs can be maintained under pathological conditions (*Figure 2—figure supplement 2*). Hence, we performed further analysis of the pattern of hair cell loss using a standardized template of cell positions.

To simplify the treatment of hair cell position in the subsequent analysis, the Cartesian coordinates of hair cell positions were transformed and normalized to match the standardized template, which consisted of the normalized two-dimensional grids parallel to the surface of the sensory epithelium (*Figure 2F,G*). The first axis was defined by the line of detected IHCs, and the second was set perpendicular to the first axis. This simplified presentation is useful for measuring the space unoccupied by the hair cells. We hypothesized that the area of the unoccupied space reflects the space previously occupied by hair cells that were subsequently lost. By dividing the areas that were not occupied by existing OHCs by the average area of a single OHC, we could estimate the number of OHCs lost. Comparison of the performances of trained operators and automated calculation confirmed that adequate estimation of hair cell loss could be achieved by automated calculation; indeed, the two methods were similarly effective (*Figure 2G* and *Table 3*). The total number of lost OHCs was 26.3 ± 6.3, 34.6 ± 5.1, 55.8 ± 4.5, and 49.3 ± 17.3 (mean ±SD) in wild-type C57BL/6J mice at PND 30, 60, and 120, and PND 60 plus noise exposure. These data are consistent with previous estimates of hair cell loss based on manual counting in rats and chinchillas (*Hu et al., 2012*; *Yang et al., 2004*). Therefore, our protocol is suitable for quantitative analysis of IHCs and OHCs, including detection and counting of lost hair cells.

## Spatial characteristics of hair cell loss

Presentation of lost cell density in the form of two-dimensional grids facilitates side-by-side comparison of hair cell loss along the longitudinal axis of the organ of Corti (*Figure 3A* and *Figure 3—figure supplement 1*). In samples from young mice not exposed to noise, small numbers of hair cells were lost along both the longitudinal and radial axes (*Figure 3A* and *Figure 3—figure supplement 1*). Samples of aged mice had a higher density of lost cells at both ends of the organ of Corti

**Table 3.** Inter-operator percent match in void space detection (related to Experimental procedures).

| Sample number | Inter-operator percent match | | | | Number of detected void space | | |
|---|---|---|---|---|---|---|---|
| | A[¶]-B[¶] | B[¶]-C[¶] | A[¶]-C[¶] | Auto[††]-HC[‡‡] | Both | Auto[††]-only | HC[‡‡]-only |
| 1[*] | 0.960 | 0.880 | 0.917 | 0.920 | 24 | 1 | 1 |
| 2[†] | 0.898 | 0.917 | 0.906 | 0.952 | 84 | 2 | 2 |
| 3[‡] | 0.923 | 0.885 | 0.958 | 0.889 | 24 | 3 | 0 |
| 4[§] | 0.923 | 0.882 | 0.846 | 0.926 | 50 | 3 | 1 |
| Overall | 0.916 | 0.898 | 0.897 | 0.931 | 182 | 9 | 4 |

*. Sample 1, two months old, total loss rate of OHCs: 1.7%.

†. Sample 2, two months old with noise exposure, total loss rate of OHCs: 8.1%.

‡. Sample 3, one month old, total loss rate of OHCs: 2.2%.

§. Sample 4, four months old, total loss rate of OHCs: 4.2%.

¶. Skilled human operators (A, B, and C).

††. Auto, automated OHC loss counting program.

‡‡. HC, human consensus.

DOI: https://doi.org/10.7554/eLife.40946.012

(*Figure 3B*). The difference between adult and aged mice was confirmed by comparison across ages. In particular, the age-dependent increase in lost cell density was prominent at the apical end (PND 60: $0.0596 \pm 0.0048$, PND 120: $0.116 \pm 0.0150$, Welch's $t$-test, $p < 0.01$, $t = 3.59$, $df = 9.68$). By contrast, previous studies reported a higher rate of ACL in the basal portion, but failed to detect a prominent increase in the proportion of lost cells in the apical region. This difference may be due to the fact that our tissue clearing technique enabled complete visualization of hair cells at the helicotrema (*Figure 3C*). ACL also had spatial features along the radial axis, with a higher density at positions distal to the modiolus (*Figure 3D*). This trend along the radial axis was already present in cochleae at PND 60, indicating that ACL may represent acceleration of a pathology already present in the early stage of life. In summary, the method we developed was well suited for comprehensive analysis of ACL.

The cellular pathology of NCL was more complex than that of ACL, exhibiting a highly variable pattern among samples. This may be inevitable in our paradigm of NCL, because this protocol is expected to induce milder insults to the sensory epithelium (*Figure 3A* and *Figure 3—figure supplement 1*). Our comprehensive analysis was useful in detecting higher variability of cell loss at the basal end after noise exposure (position 'a' against 'e' in *Figure 3*; $p < 0.001$, $F(6,9) = 15.5$) and also in aged mice, (position 'a' against 'e' in *Figure 3*; $p < 0.05$, $F(8, 8) = 5.69$), suggesting that vulnerability at the basal end may be intrinsically variable. Principal component analysis applied to the spatial pattern of cell loss was helpful in isolating ACL- and NCL-related parameters (*Figure 3—figure supplement 1*), and the results revealed that NCL had a weaker impact in the apical portion. Thus, distinct mechanisms of cellular pathology may be responsible for ACL and NCL.

## Model-based analysis of hair cell loss

The positions of putative lost cells revealed spatial clustering above the level that would be expected by chance, regardless of age and the presence or absence of noise exposure (*Figure 4A*). To evaluate the spatial patterns of clustering, we constructed two distinct mechanistic models (*Figure 4B*). In the first model, cell loss occurs stochastically, but the probability increases if adjacent cells have been lost (neighborhood effect model). In the second model, the frequency of cell loss depends on adverse factors localized along the longitudinal axis of the organ of Corti (position effect model).

Fitting of the two models was comparable in samples from aged mice or after noise exposure (*Figure 4C*), suggesting the complex relationship between lost cell clustering, various hair cell pathologies, and the extent of cell loss. Therefore, we developed a two-component model in which both the neighborhood effect and the position effect induced cell loss, but with different weights. By controlling the weights of the two effects, it was possible to improve fitting to the experimental data. The combinations of the two effects yielding the best fit to the experimental data were plotted

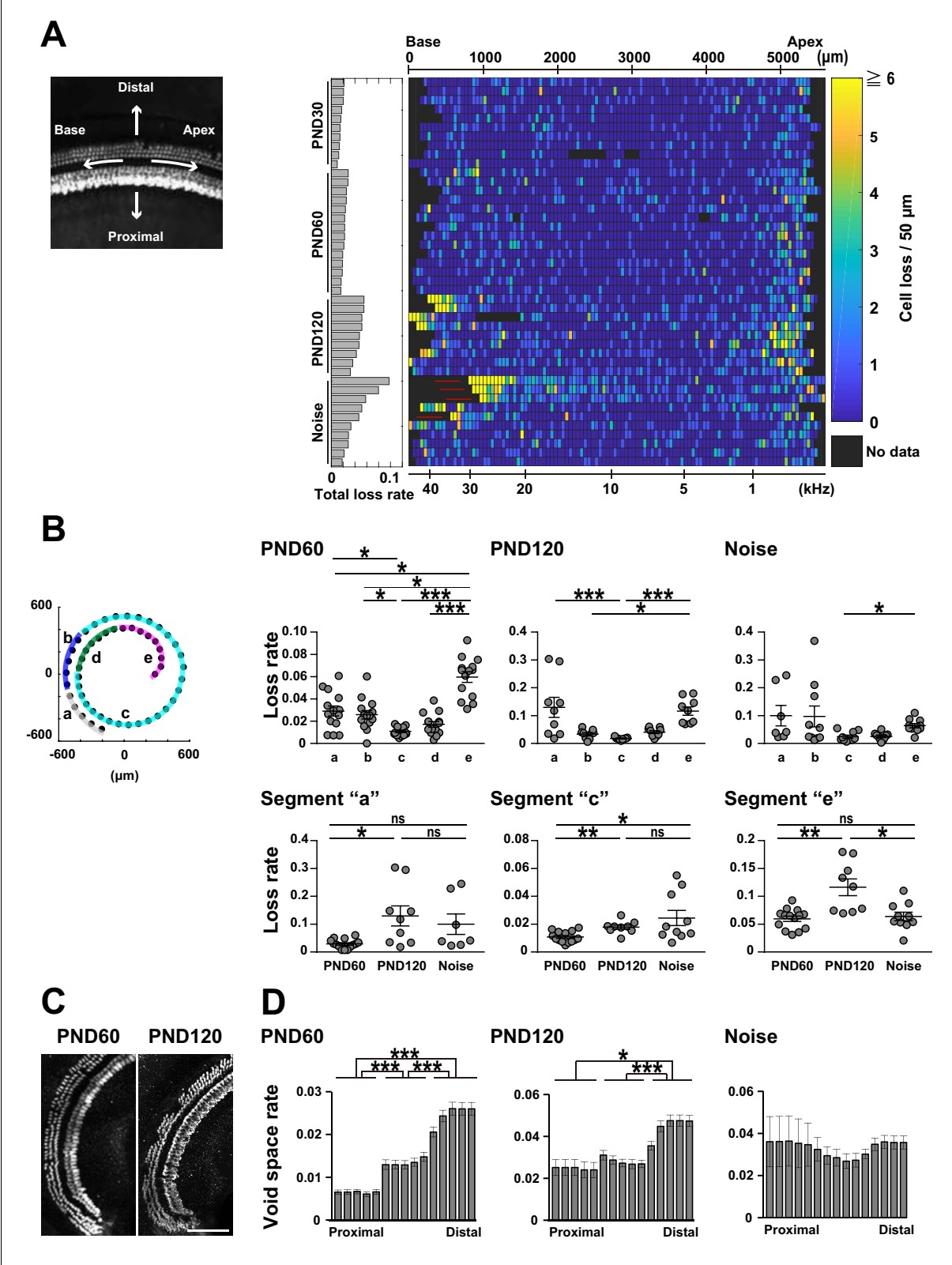

**Figure 3.** Spatial pattern of hair cell loss. (**A**) Pseudo-color presentation of hair cell loss along the longitudinal axis of the organ of Corti (PND 30, 60, and 120 and noise exposure at PND 60). Each row represents a single cochlear sample. Numbers of lost hair cells within 50 μm segments along the longitudinal axis were measured. For samples with higher cell loss in the basal portion, it was difficult to define the basal end of the sensory epithelium. These samples with ambiguous starting points of the epithelium were marked by thin red lines in rows of noise-exposed samples. The raw fluorescence

*Figure 3 continued on next page*

*Figure 3 continued*

image shows the definition of directions (distal and proximal, apex and base) relative to the sensory epithelium. (**B**) Distribution of lost cells along the longitudinal axis of the organ of Corti in three experimental groups. PND 60 and 120, and noise exposure at PND 60, exhibit distinct patterns of hair cell loss (Kruskal–Wallis test with Steel–Dwass test). (**C**) Detection of hair cell loss at the helicotrema. Scale bar, 100 µm. (**D**) Distribution of lost cells along the radial axis of the organ of Corti. Samples from PND 60 and 120 exhibit gradients of cell loss. (Paired *t*-test followed by Bonferroni's correction, *p < 0.05; ***p < 0.001.) Number of samples; n = 10 (PND 30), 14 (PND 60), 9 (PND 120), and 10 (Noise), except for n = 7 in segment 'a' of Noise in (**B**). *p < 0.05; **p < 0.01; ***p < 0.001. PND, postnatal day.

DOI: https://doi.org/10.7554/eLife.40946.013

The following source data and figure supplement are available for figure 3:

**Source data 1.** Source data for *Figure 3D* and *Figure 3—figure supplement 1*.
DOI: https://doi.org/10.7554/eLife.40946.015

**Figure supplement 1.** Longitudinal and radial distribution of hair cell loss in the organ of Corti.
DOI: https://doi.org/10.7554/eLife.40946.014

---

along with a color code for the extent of cell loss (*Figure 4D* and *Figure 4—figure supplement 1*). The overall pattern of data distribution suggests a higher contribution of the neighborhood effect in young and adult mice not exposed to noise (*Figure 4E*). With age, the contribution of the position effect increased, whereas noise exposure in adult mice resulted in a variable extent of damage; data points were dispersed, with highly damaged sensory epithelium experiencing a greater contribution from the position effect.

## Automatic evaluation of cell damage and detection of multiple intracellular components

Fluorescence-based detection of cytoskeletal components, such as F-actin, enabled us to obtain information about the integrity of subcellular structure in hair cells. We evaluated F-actin integrity of OHCs at multiple positions of the organ of Corti, specified by the extent of cell loss and clustering of lost cells (*Figure 5A*). This approach is useful for automatic evaluation of the extent of stereocilia damage at multiple points of the organ of Corti. The reduction of F-actin content in hair cells near to lost hair cells supports the neighborhood effect model of lost cell clustering, described above.

We also tried to image subcellular components in hair cells using specific antibodies against pre-synaptic ribbons (C-terminal-binding protein 2, CtBP2) and synaptic vesicles (vesicular glutamate transporter type 3, VGLUT3). Similar immunocytochemical approaches were applied to other components of the organ of Corti, including axons immunopositive for high–molecular weight neurofilament protein (NF200) and supporting cells positive for SRY (sex determining region Y)-box 2 (SOX2) immunoreactivity (*Oesterle et al., 2008*) (*Figure 5B*). These results suggest that this method can be applicable to analysis of multiple cellular components in the cochlea. In this study we utilized a water immersion objective with moderate numerical aperture (NA). In future, modification of our method with an objective lens with higher NA may enable more precise imaging of intracellular structure in a scale of the entire cochlea.

In this study, we developed a rapid method for optical tissue clearing, labeling, and automated image analysis of the inner ear. Currently available tissue clearing and labeling technologies have limited applicability to hard tissues, including bone, tooth, cartilage, and tendon. Effective removal of fine hydroxyapatite crystals in hard tissues is a key to establishing clearing methods. Here, we demonstrated that our modified Sca*l*eS method represents a powerful approach for exhaustive analysis of expression profiles in hair cells along the entire organ of Corti, using multiple antibodies. This technique can be directly applied to the characterization of genetic and environmental models of hearing loss. In future, the analytical pipeline we developed will be integrated with active elimination of bone mineral and organic components by physical principles (*Lee et al., 2016*). To further increase efficiency, the decalcification solution should be elaborated. Rapid decalcification can be achieved by combining EDTA with formic or hydrochloric acid (*Treweek et al., 2015*). A recent report also examined multiple conditions of clearing hard tissues and recommended lowering pH of the EDTA-containing decalcification solution (*Tainaka et al., 2018*). However, prolonged sample treatment with high concentrations of acid can reduce immunoreactivity and accelerate quenching of fluorescent proteins. Future investigations should seek to establish clearing and labeling methods optimized for a wide spectrum of hard tissue components.

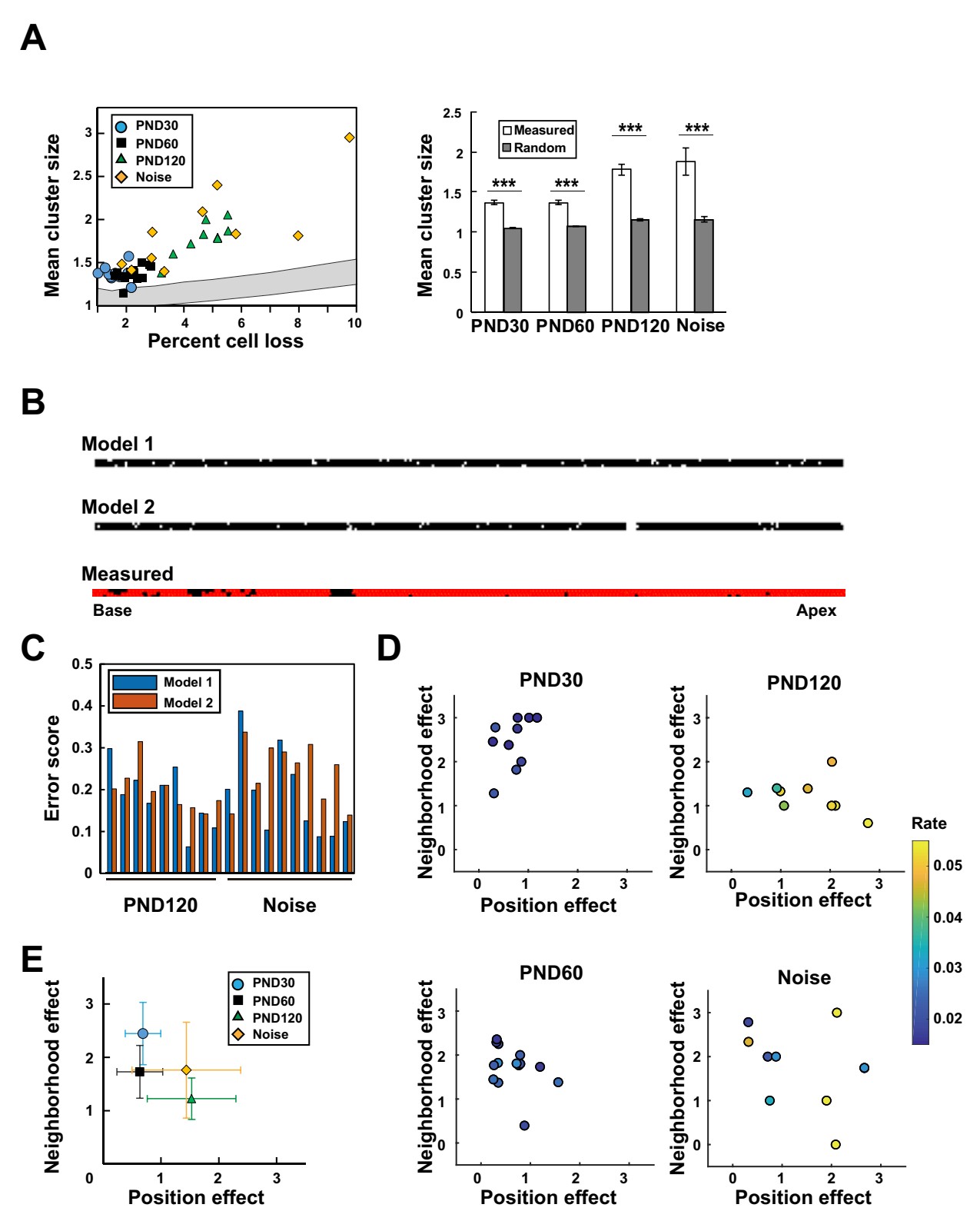

**Figure 4.** Model-based analysis of clustered cell loss. (**A**) Evaluation of the extent of clustered cell loss by comparison with the extent of clustering based on a model of random cell loss. The extent of cell clustering in the experimental data was much higher than would have been expected from random cell loss (99% confidence intervals within two lines) (Welch's *t*-test, ***p < 0.001). (**B**) Construction of two models of hair cell loss (upper: neighborhood effect model; lower: position effect model). Virtual cell loss data were generated from the models and compared with the experimental

*Figure 4 continued on next page*

*Figure 4 continued*

data (measured). (C) Evaluation of the goodness-of-fit of the two models to the experimental data using the error score, which measures the extent of deviation of the clustering properties generated by the models from those observed in real samples. (D) Assessment of the relative contributions of the two models (neighborhood effect and position effect) to achieve the best fit to the experimental data. The two models contribute differentially under various conditions. The color code shows the proportion of lost OHCs against the total OHCs. (E) Overall pattern of contribution from two models. Note that the neighborhood effect makes a stronger contribution in young adult mice, whereas the position effect makes a stronger contribution in aged mice (means ± SD). Number of samples; n = 10 (PND 30), 14 (PND 60), 9 (PND 120), and 10 (Noise). PND, postnatal day.

DOI: https://doi.org/10.7554/eLife.40946.016

The following source data and figure supplement are available for figure 4:

**Source data 1.** Source data for *Figure 4A, C, D and E*.
DOI: https://doi.org/10.7554/eLife.40946.018
**Figure supplement 1.** Simulation analysis of clustered cell loss.
DOI: https://doi.org/10.7554/eLife.40946.017

System-level analysis of the organ of Corti is important for extracting the operating principles of mechanosensory transduction. In parallel, generation of a variety of model mice harboring mutations in genes involved in hearing function will facilitate functional studies. While the functional consequences of gene mutation can be assessed using standardized protocols, such as ABR, at present we have no widely approved format for the assessment of cellular pathology. The method we developed in this study may be useful for standardization of cell-based analysis. A recent study of in situ two-photon imaging of the organ of Corti revealed the detailed architecture of the mechanical framework in the sensory epithelium (*Soons et al., 2015*). The method described here could be combined with information about mechanical characteristics. By integrating position-specific mechanical property, fluid dynamics, and hair cell physiology, such an approach would be useful for modeling of cochlear function (*Liu et al., 2015*). Manual identification of more than 2500 hair cells per sample and subsequent analysis of cell loss is not possible for large sets of cleared samples from animals of different ages, genetic backgrounds, and experimental conditions. Accordingly, the analytical pipeline described here was designed to minimize manual processing. Objective comparison of position-dependent cell pathology among multiple mouse models of hearing loss will facilitate identification of critical molecular signatures associated with cochlear pathology.

## Materials and methods

For detailed procedures, see Appendix 1 and 2.

**Key resources table**

| Reagent type (species) or resource | Designation | Source or reference | Identifiers | Additional information |
|---|---|---|---|---|
| Genetic reagent (M. musculus) | C57BL/6J | Sankyo Lab (JAPAN) | PRID:MGI:5658686 | |
| Genetic reagent (M. musculus) | CBA/Ca | Sankyo Lab (JAPAN) | PRID:MGI:2159826 | |
| Genetic reagent (M. musculus) | Thy1-GFP line-M | Jackson Lab | PRID:MGI: 3766828 | |
| Genetic reagent (M. musculus) | GO-Ateam | PMID: 19720993 | | Dr. M Yamamoto (Kyoto University, Japan) |
| Antibody | Rabbit polyclonal anti-Myosin VIIa | Proteus Biosciences | cat# 25–6790 PRID:AB_10013626 | IHC (1:100) |

*Continued on next page*

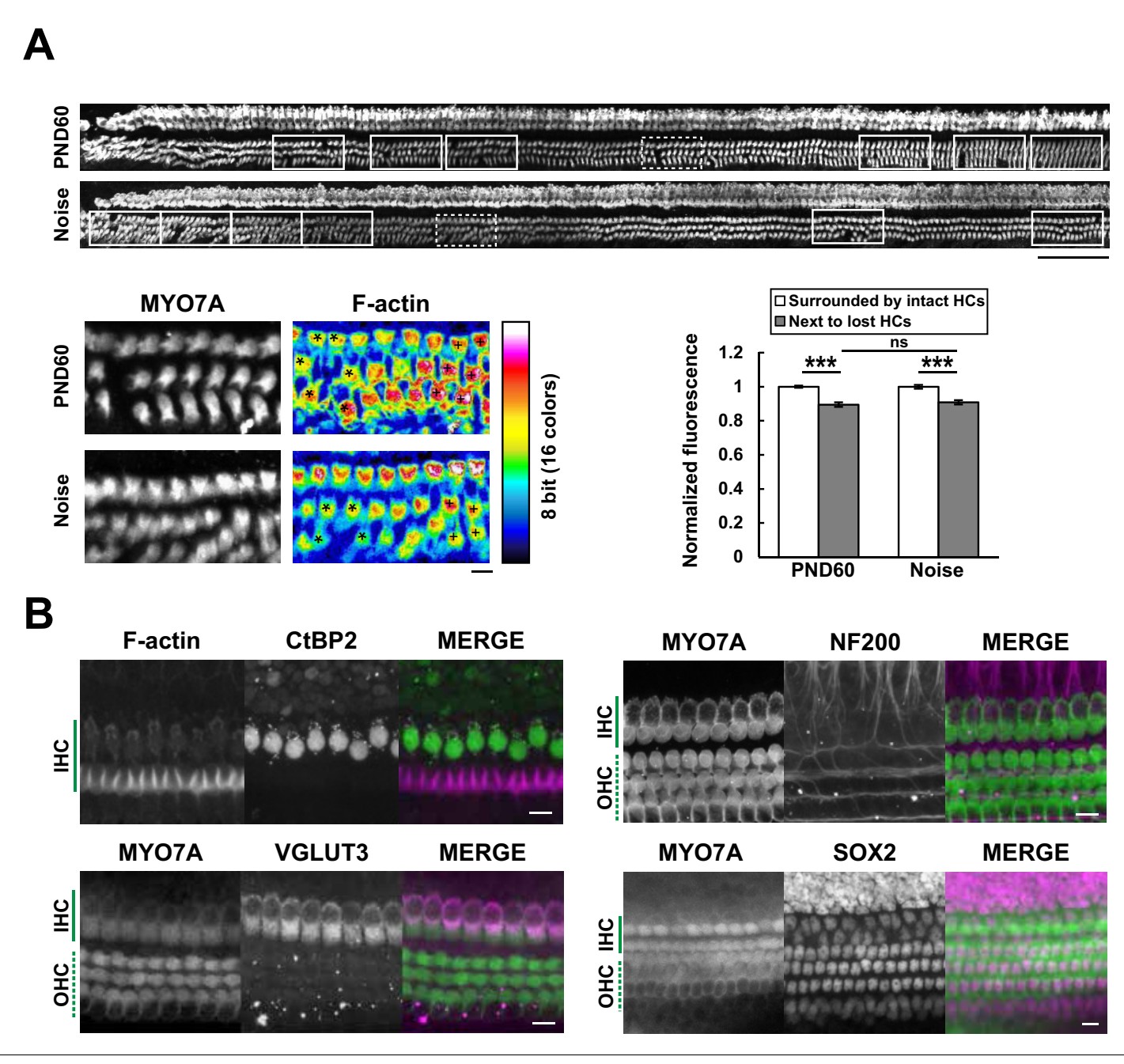

**Figure 5.** Efficient mapping of subcellular pathology and multiple cellular components. (**A**) Automated detection of areas with variable degrees of hair cell loss, combined with evaluation of subcellular pathology. All sites of hair cell loss (white squares) were selected, and changes in the F-actin content were evaluated (upper image). White squares with dotted lines are representative analysis areas, and are enlarged at lower left. Hair cells surrounded by intact hair cells (crosses) or next to lost cells (asterisks) were compared for their F-actin content (lower right). The graph reveals loss of F-actin in hair cells adjacent to lost cells. Scale bars, 100 μm (upper) and 10 μm (lower). [n = 108 (PND 60) and 103 (Noise), paired t-test for comparison within the group, Welch's t-test with Bonferroni's correction for comparison of cell groups between different experimental conditions, ***p < 0.001; ns, not significant, p > 0.05.] (**B**) Modified ScaleS technique can be adapted to multiple immunohistochemistry of cellular and subcellular components at PND 5. Antibodies against CtBP2, VGLUT3, NF200, and SOX2 were used to detect multiple components in situ. Scale bars, 10 μm. HC, hair cell; IHC, inner hair cell; OHC, outer hair cell; PND, postnatal day.

DOI: https://doi.org/10.7554/eLife.40946.019

The following source data is available for figure 5:

**Source data 1.** Source data for *Figure 5A*.
DOI: https://doi.org/10.7554/eLife.40946.020

*Continued*

| Reagent type (species) or resource | Designation | Source or reference | Identifiers | Additional information |
|---|---|---|---|---|
| Antibody | Mouse monoclonal anti-Neurofilament 200 | SIGMA | cat# N5389 PRID:AB_260781 | IHC (1:100) |
| Antibody | Mouse monoclonal anti-SOX-2 | EMD Millipore | cat# MAB4343 PRID:AB_827493 | IHC (1:200) |
| Antibody | Mouse monoclonal anti-CTBP2 | BD Bioscience | cat# 612044 PRID:AB_399431 | IHC (1:100) |
| Antibody | Guinea pig polyclonal anti-VGLUT3 | PMID: 20034056 | | IHC (1:500), Dr. H Hioki (Juntendo University, Japan) |
| Antibody | Alexa Fluor 488-conjugated mouse monoclonal anti-VE cadherin | eBioscience | cat# 16-1441-81 PRID:AB_15604224 | IHC (1:500) |
| Chemical compound, drug | Rhodamine phalloidin | Invitrogen | cat# R415 | IHC (1:500) |
| Chemical compound, drug | Triton X-100 | Nakalai-tesque | cat# 12967–45 | |
| Chemical compound, drug | Urea | SIGMA | cat# U0631-1KG | |
| Chemical compound, drug | N,N,N′,N′-Tetrakis (2-eydroxypropyl) ethylendiamine | TCI | cat# T0781 | |
| Chemical compound, drug | D-sucrose | Wako | cat# 196–00015 | |
| Chemical compound, drug | 2,2′,2′′-nitrilotriethanol | Wako | cat# 145–05605 | |
| Chemical compound, drug | Dichloromethane | SIGMA | cat# 270997–100 ML | |
| Chemical compound, drug | Tetrahydrofuran | SIGMA | cat# 186562–100 ML | |
| Chemical compound, drug | Dibenzyl Ether | Wako | cat# 022–01466 | |
| Chemical compound, drug | Methanol | Wako | cat# 132–06471 | |
| Chemical compound, drug | D-glucose | SIGMA | cat# G8270-100G | |
| Chemical compound, drug | D-sorbitol | SIGMA | cat# S1816-1KG | |
| Chemical compound, drug | Thiodiethanol | Wako | cat# 205–00936 | |
| Chemical compound, drug | Acrylamide | Wako | cat# 011–08015 | |
| Chemical compound, drug | Bis-acrylamide | SIGMA | cat# 146072–100G | |
| Chemical compound, drug | VA-044 initiator | Wako | cat# 225–02111 | |
| Chemical compound, drug | Sodium dodecyl sulfate | TCI | cat# I0352 | |
| Chemical compound, drug | FocusClear | CelExplorer Labs | cat# F101-KIT | |
| Chemical compound, drug | Glycerol | Wako | cat# 075–00616 | |
| Chemical compound, drug | Dimethyl sulfoxide | Wako | cat# 043–07216 | |

*Continued on next page*

*Continued*

| Reagent type (species) or resource | Designation | Source or reference | Identifiers | Additional information |
|---|---|---|---|---|
| Chemical compound, drug | N-acetyl-L-hydroxyproline | TCI | cat# A2265 | |
| Chemical compound, drug | Methyl-β-cyclodextrin | TCI | cat# M1356 | |
| Chemical compound, drug | γ-cyclodextrin | TCI | cat# C0869 | |
| Chemical compound, drug | Tween-20 | Wako | cat# 167–11515 | |
| Software, algorithm | ImageJ | NIH | PRID: SCR_003070 | |
| Software, algorithm | GraphPad Prism 6 | GraphPad Software | PRID: SCR_002798 | |
| Software, algorithm | MATLAB | MathWorks | PRID: SCR_001622 | |
| Software, algorithm | Microsoft Excel | Microsoft | PRID: SCR_016137 | |
| Software, algorithm | Adobe Illustrator | Adobe | PRID: SCR_010279 | |
| Software, algorithm | Signal processor | Nihon Kouden | Neuropack MEB2208 | |
| Other | MATLAB codes | This paper | | https://github.com/okabe-lab/cochlea-analyzer |
| Other | 25x water-immersion objective lens | Nikon | N25X-APO-MP | |
| Other | 25x water-immersion objective lens | Olympus | XPLN25XWMP | |
| Other | Sound speaker | TOA | HDF-261–8 | |
| Other | Power amplifier | TOA | IP-600D | |
| Other | Condenser microphone | RION | UC-31 and UN14 | |
| Other | Sound calibrator | RION | NC-74 | |
| Other | Noise generator | RION | AA-61B | |
| Other | Dual channel programmable filter | NF corporation | 3624 | |

## Tissue acquisition

After euthanasia, mice were perfused transcardially with 4% paraformaldehyde in PBS. Osteochondral samples (cochlea embedded in temporal bones and femurs) and other soft tissues (brain, heart, stomach, lung, liver, kidney, intestine, and spleen) were isolated by standard dissection techniques.

## Decalcification

Samples were washed for 30 to 180 min in PBS containing 0.1% Triton X-100 with continuous rocking at 40 rpm. Decalcification was performed by incubating samples for 48 to 120 hr in 500 mM EDTA in PBS at 37°C, and terminated by washing samples several times with PBS.

## Tissue extraction

Samples were placed in a solution containing 3 M guanidinium chloride, 35% (w/v) D-sorbitol, 15% (w/v) D-glucose, and 4% (w/v) Triton X-100 in PBS (pH 6.0–8.0) and incubated at 37°C for 2 to 12 hr.

## Labeling with antibodies and small molecules

After tissue extraction, samples were washed with PBS containing 0.1% Triton X-100 for 30 min with continuous rocking at 40 rpm. Samples were incubated for 2 to 48 hr in a solution containing primary antibodies or small molecules (details provided in Appendix 1) with appropriate dilutions at 37°C. Unbound antibodies or small molecules were removed by washing for 30 min with PBS containing 0.1% Triton X-100, with continuous rocking at 40 rpm. Primary antibodies were detected by incubation for 12 to 48 hr with a solution containing secondary antibodies at 37°C, followed by washing as described for removal of primary antibodies. Duration of antibody incubation was adjusted depending on the size of the sample and the affinity and specificity of the antibodies.

## Adjustment of RI

For the adjustment of tissue RI, samples were incubated for 15 min to 2 hr at 37°C in a RI matching solution. The duration of this step was adjusted depending on the size and properties of the sample. Our optimized RI matching solution (RI = 1.47) contained 3 M guanidinium chloride (or 4 M urea), 60% (w/v) D-sorbitol, and 0.1% (w/v) Triton X-100 in PBS (pH 7.1). For the optimization of RI, we tested multiple RI matching solutions with their RIs ranging from 1.41 to 1.56. The RI matching solutions with their RI lower than 1.47 were made by diluting the RI matching solution with RI = 1.47 with water. The final RIs were confirmed by a refractometer. The RI matching solutions with RI = 1.52 and 1.57 were thiodiethanol and dibenzyl ether, respectively. After RI adjustment, samples were placed in a chamber with the same RI matching solution, covered by a coverslip, and imaged by a two-photon microscope. The same cochlear sample was imaged repetitively in the RI matching solutions with increasing RIs. The maximal image depth was determined by measuring the distance from the bone surface to the deepest position where fluorescence signal of MYO7A-positive hair cells can be detected (*Figure 1—figure supplement 1*). In total, five independent cochlear preparations were imaged.

## Microscopy and image acquisition

Imaging of IHCs and OHCs of the organ of Corti was performed on a two-photon microscope (Nikon A1MP) equipped with a mode-locked Ti:sapphire laser (Mai Tai Deep See, Spectra Physics) operated at 800 nm with a 25 × water immersion objective lens (NA = 1.10). A chamber containing the sample was filled with the RI matching solution, covered by a glass coverslip, and placed under the objective lens. The size of single horizontal images was set to 512 × 512, with pixel sizes of 0.99 × 0.99 μm and z-spacing of 1 μm. Images were successively acquired with 10–40% overlap. Image processing was performed using the ImageJ software (National Institute of Health), and three-dimensional rotation was performed using Imaris (Bitplane), FluoRender (Version 2.18, the University of Utah), and NIS-Element AR (Version 4.51, Nikon). Adjustment of fluorescence intensity along the longitudinal axis of the organ of Corti was performed using a MATLAB script written in-house (MathWorks).

## Automated cell-count and three-dimensional morphology analysis

Hair cell detection and analysis were performed automatically using custom MATLAB scripts (R2017b, MathWorks); details are provided in Appendix 2. MATLAB source code is available on GitHub (*Iida, 2018a*; copy archived at https://github.com/elifesciences-publications/cochlea-analyzer).

### Step 1: Stitching of multiple image stacks into a single stack

Multiple image stacks containing portions of the organ of Corti were assembled into a single image stack. Shifts of coordinates between image stacks were calculated based on cross-correlation (MATLAB 'normxcorr2' function). After image stitching, a blending algorithm (*Rankov et al., 2005*) was applied to remove sharp intensity changes in the zone of overlap.

### Step 2: Reconstruction of linearized image

Hair cells in each image stack were detected as local intensity peaks (MATLAB 'imregionalmax' function). Single-linkage clustering (maximal distance of connection, 25 μm) was effective for eliminating or reducing the number of false positives. A stretch of local peaks corresponding to the entire row of hair cells were divided into segments of 200–300 μm in length. In each segment, the best-fit plane

was calculated (MATLAB 'pca' function), together with the best-fit arc along the rows of hair cells. The multiple best-fit arcs were stitched into a continuous curve (*Figure 2B*). A voxel image containing the entire straightened row of hair cells was reconstructed from the image stacks, based on the stitched-fit curve and the normal vectors of the fit planes.

### Step 3: Automated detection of IHCs

First, local correlation between the hair cell template and the voxel image of linearized epithelium was calculated by template matching (MATLAB 'normxcorr2' function), and the peaks of the correlation were detected. Pixels corresponding to detected peaks were grouped according to the physical size of the IHCs via connected-component labeling. These connected pixel groups (hereinafter called 'cell candidates') were used as a first approximation of IHC positions linked to other attributes, including correlation values and local intensity distributions.

The cell candidates were further evaluated to eliminate false positives using two successive machine learning models. The first ensemble learning method created the model for selection with predictor data consisting of areas, barycentric coordinates, correlation values, the intensities of the peaks, and the corresponding values of nearby cell candidates (MATLAB 'fitensemble' function with 'GentleBoost' method) (*Friedman et al., 2000*). The model was trained to calculate posterior probability (prediction score), and cell candidates with a high prediction score (~1000 candidates out of initial ~50,000) were selected and further analyzed by the second ensemble learning method (MATLAB 'fitensemble' function with 'Bag' method) (*Breiman, 2001*). This method was based on expanded predictors (the prediction score from the first step of the candidate and nearby candidates, and the local intensity distribution centered on the barycentric coordinates of the peaks). The cell candidates after the second selection were connected sequentially, subject to the physical constraint that the IHCs must form a single row with roughly constant intervals of more than 6 μm. The resulting putative positions of IHCs were used for fine readjustment of image linearization and three-dimensional structural analysis.

### Step 4: Automated detection of OHCs

The image processing applied for IHCs in Step 3 was also applied to OHCs. Detection accuracy was improved by two additional evaluations based on machine learning. First, physical constraints of OHC alignment were introduced into three rows. A multiclass classification model, based on the convolutional neural network method [Neural Network Toolbox of MATLAB (*LeCun et al., 1989*)], sorted cell candidates into respective rows using input images each containing three rows of four or five OHCs. If the distance between two adjacent cell candidates in the same row exceeded 1.5 times the average distance, the presence of additional cells in the gap was assessed by the fourth model based on the convolutional neural network method. Input images for machine learning were sampled by placing small rectangular areas at equal distances from one another within the gap. If the model predicted the existence of additional cells in the gap, the nearest peaks of the correlation coefficient from the first template matching were recovered.

## Frameworks of machine learning models

Details of the models used in the detection are shown in *Table 1* and *Table 4* (*Sokolova and Lapalme, 2009*). Ensemble learning methods were applied to a one-dimensional predictor data set, and the convolutional neural network method was applied to a two-dimensional predictor data set (images). For the first ensemble learning in Steps 3 and 4, the GentleBoost algorithm was selected because of its superior training performance on large data sets relative to the Random Forest algorithm [GentleBoost, MATLAB 'fitensemble' function with 'GentleBoost' method (*Friedman et al., 2000*); Random Forest, 'Bag' method in MATLAB (*Breiman, 2001*)].

## Evaluation of automated detection system for hair cells

The detection efficiency of the system is shown in *Table 2*. The models used in the system were trained on ten cochleae as described above, and the efficiency of the trained system was evaluated on ten other cochleae. The results of auto-detection were compared against a reference created by independent manual counting by three human operators. The reference contains fluorescent objects judged to be hair cells by at least two operators.

**Table 4.** Number of training and test dataset, and performance evaluation of machine learning models (related to **Figure 2**).

| Model | Training | | Test | | Recall | Precision | F score |
|---|---|---|---|---|---|---|---|
| | Total (n) | Positive labels (n) | Total (n) | Positive labels (n) | | | |
| IHC* 1 | 607,954 | 5906 | 578,851 | 5741 | 0.961 | 0.941 | 0.951 |
| IHC* 2 | 37,576 | 11,977 | 18,104 | 5753 | 0.977 | 0.986 | 0.981 |
| OHC† 1 | 1,112,659 | 20,576 | 1,099,519 | 19,959 | 0.978 | 0.914 | 0.945 |
| OHC† 2 | 28,702 | 20,576 | 27,185 | 19,959 | 0.959 | 0.979 | 0.969 |
| OHC† 3 | 20,416 | Row1: 6706 Row2: 6745 Row3: 6965 | 19,594 | Row1: 6421 Row2: 6450 Row3: 6723 | 0.993‡ | 0.993‡ | 0.993‡ |
| OHC† 4 | 4114 | 1365 | 2990 | 905 | 0.920 | 0.946 | 0.933 |

*. IHC, inner hair cell.

†. OHC, outer hair cell.

‡. Calculated by micro-average of recall and precision (Sokolova M and Lapalme G, 2009)

DOI: https://doi.org/10.7554/eLife.40946.021

### Analysis of spatial distribution of OHCs

Loss of OHCs results in formation of spatial gaps. To evaluate the extent of cell loss, conventional manual counting estimates the number of lost cells based on the sizes of spatial gaps. In this study, a method that can directly and systematically evaluate the sizes of holes without assuming horizontal rows of hair cells was introduced. The first step of this method was equalization of the coordinates of detected cells throughout the cochlear. Cell positions were adjusted to normalize the average intercellular distance both horizontally and vertically, and to normalize the intercellular distances along the entire organ of Corti. Subsequent placement of square areas with positions matched to the normalized coordinates of detected hair cells left connected pixel groups corresponding to the spaces of putative lost cells. Details of these analyses are provided in Appendix 2.

### Principal component analysis on OHC loss frequency

Principal component analysis was performed on the OHC loss frequency along the longitudinal and radial axes of NCL and ACL samples. Variables were the frequency of OHC loss in specific spatial segments. These spatial segments were 13 longitudinal and 15 radial segments that equally divide the total area. A singular value decomposition algorithm was utilized for the calculation of coefficients for the first and second principal components ('svd' option of MATLAB 'pca' function).

### Analysis of the three-dimensional structure of the cochlea

The spiral structure of the cochlea was analyzed based on the three-dimensional spatial distribution of IHCs because these cells formed a row that was rarely disturbed. Details of these analyses are provided in Appendix 2.

### Simulation analysis of clustered cell loss

It was observed that lost OHCs tended to be clustered. Simulation analysis was performed to evaluate two independent factors that could be responsible for such clustering. (1) A lost cell increases the probability that neighboring cells will be lost (Model 1; neighborhood effect). (2) Cell loss takes place with a probability that is a function of the local environment of the sensory epithelium (Model 2; position effect). The simulation was performed on each cochlea sample, using the measured ratio of cell loss, the number of clusters, and the cluster sizes. The simulation was performed on two matrices, the 'cell matrix' and 'probability matrix', with sizes of 3 rows × 600 columns corresponding to the distribution of OHCs (*Figure 4—figure supplement 1*). The cell matrix recorded the positions of cell loss, and the probability matrix recorded the probabilities of cell loss in each step of the simulation.

Each operation started with a cell matrix with no lost cells and a probability matrix with or without an initial position effect. In each step, a single cell was selected for removal with a probability given

by the probability matrix, and the position was recorded in the cell matrix. The operation was stopped when the total number of lost cells reached the number of cells lost in a given sample.

The neighborhood effect was created by adding an additional weight to the adjacent probability matrix elements. To introduce the position effect, random numbers were generated according to a power-law distribution calculated by the following function.

$$P(x) = 0.1 \times p \times X^{(1-0.1 \times p)},$$

where p is the parameter controlling the shape of distribution. The values along the row of the probability matrix were obtained from the function P(x), with input x drawn from a uniform distribution between 0 and 1. Values in the same column were set to be identical. Gaussian filtering was applied to the probability matrix to broaden the peak width.

A panel of 16 simulated histograms was created for each sample by changing the relative weights of two effects (neighborhood and position effects) (*Figure 4—figure supplement 1*). The operation was repeated 500 times for each parameter set. A histogram of cluster size was constructed, and similarity to the measured data was evaluated (*Figure 4—figure supplement 1*). The extent of histogram dissimilarity between measured and simulation results was calculated by the sum of squared errors (error score), and a two-dimensional heat map was created (*Figure 4—figure supplement 1*). The weighted average of the top three combinations of weights was calculated and taken to represent the relative contribution of two factors to cell loss events.

## Acknowledgments
We thank Dr. Hiroyuki Hioki for anti-VGLUT3 antibody.

## Additional information

### Funding

| Funder | Grant reference number | Author |
| --- | --- | --- |
| Ministry of Education, Culture, Sports, Science, and Technology | 26111506 | Chisato Fujimoto |
| Japan Society for the Promotion of Science | 15K10743 | Chisato Fujimoto |
| Japan Society for the Promotion of Science | 26253081 | Tatsuya Yamasoba |
| Japan Society for the Promotion of Science | 16K15717 | Tatsuya Yamasoba |
| Japan Science and Technology Agency | JPMJCR14W2 | Shigeo Okabe |
| Japan Agency for Medical Research and Development | 17gm5010003 | Shigeo Okabe |
| Japan Society for the Promotion of Science | 17H01387 | Shigeo Okabe |
| The UTokyo Center for Integrative Science of Human Behavior | | Shigeo Okabe |
| Ministry of Education, Culture, Sports, Science and Technology | 18H04727 | Shigeo Okabe |

The funders had no role in study design, data collection and interpretation, or the decision to submit the work for publication.

## Author contributions
Shinji Urata, Conceptualization, Data curation, Formal analysis, Validation, Investigation, Visualization, Methodology, Writing—original draft; Tadatsune Iida, Conceptualization, Data curation, Software, Formal analysis, Validation, Investigation, Visualization, Methodology, Writing—original draft; Masamichi Yamamoto, Resources, Writing—original draft; Yu Mizushima, Formal analysis, Investigation; Chisato Fujimoto, Supervision, Funding acquisition, Writing—review and editing; Yu Matsumoto, Supervision, Writing—review and editing; Tatsuya Yamasoba, Conceptualization, Supervision, Funding acquisition, Project administration, Writing—review and editing; Shigeo Okabe, Conceptualization, Supervision, Funding acquisition, Writing—original draft, Project administration

## Author ORCIDs
Shinji Urata ⓘ http://orcid.org/0000-0002-5947-6842
Yu Mizushima ⓘ http://orcid.org/0000-0002-8184-4437
Shigeo Okabe ⓘ http://orcid.org/0000-0003-1216-8890

## Ethics
Animal experimentation: This study was performed in strict accordance with the recommendations in the University of Tokyo. All of the animals were handled according to approved institutional animal care and use committee protocol (Medicine-P12-138) of the University of Tokyo.

## Decision letter and Author response
Decision letter https://doi.org/10.7554/eLife.40946.039
Author response https://doi.org/10.7554/eLife.40946.040

# Additional files

## Supplementary files
• Transparent reporting form
DOI: https://doi.org/10.7554/eLife.40946.022

## Data availability
Data generated or analysed during this study are included in the supporting files (Figures 1-5 - Source Data, excel files). Source code is available on GitHub at https://github.com/okabe-lab/cochlea-analyzer and https://github.com/okabe-lab/Watershed

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

# Appendix 1

DOI: https://doi.org/10.7554/eLife.40946.023

## Supplemental materials and methods

### Animals

Mouse husbandry, anesthesia, and euthanasia conformed to related regulations of the government and the institutional guidelines. Protocols related to animal handling were approved by the Animal Care and Use Committee of the Graduate School of Medicine, the University of Tokyo. Male or female wild type C57BL/6J, ICR, and CBA/Ca mice at ages of PND 0 to 360 were used for application of modified Sca/eS with or without antibody labeling. For the detection of fluorescent protein signals, Thy1-GFP M line and a transgenic line expressing a fluorescence resonance energy transfer (FRET)-based indicators (GO-ATeam mouse line) (*Imamura et al., 2009*; *Nakano et al., 2011*) were used at ages of PND 0 to 120.

### Tissue acquisition

After euthanasia, mice were perfused transcardially with 4% paraformaldehyde (PFA) in phosphate buffered saline (PBS). Osteochondral samples (cochlea embedded in temporal bones and femurs) and other soft tissues (brain, heart, stomach, lung, liver, kidney, intestine, and spleen) were isolated by standard dissection techniques. For mice younger than PND 5, the step of transcardial perfusion was omitted. The isolated tissues were placed in the same fixative (4% PFA in PBS) for 10 to 24 hr. Fixed brain samples were sectioned into 100 µm to 2 mm thick sections using a vibratome.

### Tissue clearing methods

**iDISCO**; iDISCO was performed according to the published protocol (*Renier et al., 2014*). Fixed samples were treated with methanol with increasing concentrations, bleached with 5% $H_2O_2$, and rehydrated by decreasing concentration of methanol in PBS. Anti-MYO7A staining was performed after 0.3 M glycine treatment and blocking with 6% normal goat serum. Tissue clearing was achieved by treatment with increasing concentration of tetrahydrofuran/$H_2O$ up to 80% (v/v), subsequent transfer to dichloromethane, and final incubation with dibenzyl ether.

 **3DISCO**; 3DISCO was performed according to the published protocols (*Acar et al., 2015*; *Ertürk et al., 2012*; *Yokomizo et al., 2012*). Fixed samples were blocked with normal goat serum and then reacted with primary and secondary antibodies. After immunolabeling, samples were treated with increasing concentration of tetrahydrofuran/$H_2O$ up to 80% (v/v), subsequent transfer to dichloromethane, and final incubation with dibenzyl ether.

 **CLARITY**; CLARITY was performed according to the published protocols (*Chung et al., 2013*; *Tomer et al., 2014*). Briefly, animals were fixed by perfusion with a solution containing 4% paraformaldehyde, 4% acrylamide, 0.05% bis-acrylamide, 0.25% VA-044 initiator in PBS, followed by induction of hydrogel tissue embedding by raising temperature to 37°C for 3 hr. After electrophoretic tissue clearing in a chamber containing 200 mM boric acid and 4% sodium dodecyl sulfate with 20 V applied across the sample at 37°C for 72 hr, samples were blocked with normal goat serum, reacted with primary and secondary antibodies. After washing, the samples were placed in FocusClear (CelExplorer Labs) medium.

 **CUBIC**; CUBIC was performed according to the published protocols (*Susaki et al., 2014*; *Susaki et al., 2015*; *Tainaka et al., 2014*). Briefly, fixed samples were placed in ScaleCUBIC-1 (25% urea, 25% N,N,N',N'-tetrakis (2-hydroxypropyl) ethylenediamine, 15% Triton X-100) at 37°C for 24 hr, followed by brief washing and incubation with Sca*le*CUBIC-2 (50% sucrose, 25% urea, 10% 2,2',2''-nitrilotriethanol) for 2 hr. After brief washing, samples were placed in Sca*le*CUBIC-2.

**CB-perfusion**; CB-perfusion was performed according to the published protocols (*Tainaka et al., 2014*). Briefly, animals were first treated with perfusion of an excess amount of 4% paraformaldehyde in PBS, followed by sequential perfusion with PBS and 50% of ScaleCUBIC-1. The target organs were excised and immersed in ScaleCUBIC-1 for 5 days. After this step, samples were transferred to ScaleCUBIC-2 and incubated for 2–3 days. After brief washing, samples were placed in ScaleCUBIC-2.

**ScaleS**; ScaleS was performed according to the published protocols (*Hama et al., 2015*). Briefly, inner ear was isolated by a standard dissection procedure. Cochleae were isolated and 4% paraformaldehyde in PBS was perfused from the oval window. ScaleS0 (20% sorbitol, 5% glycerol, 3% dimethylsulfocide, 1% N-acetyl-L-hydroxyproline, 1 nM methyl-$\beta$-cyclodextrin, 1 mM $\gamma$-cyclodextrin), ScaleA2 (10% glycerol, 4 M urea, 0.1% Triton-X-100), ScaleB4(0) (8 M urea), and ScaleA2 were applied sequentially for permeabilization and tissue clearing. After samples were returned to PBS, immunolabeling was performed. Samples were rinsed by AbScale solution (0.33M urea, 0.1–0.5% Triton X-100 in PBS) and AbScale rinse solution (2.5% BSA, 0.05% Tween-20 in 0.1 × PBS). Finally, samples were placed in ScaleS4 (40% sorbitol, 10% glycerol, 4 M urea, 0.2% Triton X-100, 15–25% dimethylsulfoxide).

## Antibodies
Antibodies used in this study were as follows:

MYO7A (rabbit polyclonal, Proteus Bioscience), 200 kDa subunit of neurofilament protein (anti-NF200, mouse monoclonal, clone NE14, Sigma Aldrich), SOX2 (rabbit polyclonal, EMD Millipore), CtBP2 (mouse monoclonal, clone 16/CTBP2, BD Biosciences), VGLUT3 (guinea pig polyclonal, kindly gifted from Dr. Hioki), Alexa Fluor 488 goat anti-rabbit IgG (H + L) (Life Technologies), Alexa Fluor 546 goat anti-mouse IgG (H + L) (Life Technologies), Alexa Fluor 647 goat anti-mouse IgG (H + L) (Life Technologies), and Alexa Fluor 488-conjugated VE cadherin (mouse monoclonal, clone BV13, eBioscience). Concentration of antibodies should be adjusted depending on the sample size. Rhodamine phalloidin (Thermo Fisher) was used for labeling F-actin.

## Confocal microscopy
For samples labeled with multiple antibodies shown in *Figure 5B*, images were obtained by a FV1000 laser scanning microscope (Olympus) with a 25 × objective lens (NA = 1.05). Fluorophores were excited by 488, 564, and 635 nm lines of diode lasers. The sizes of single horizontal images were set to 512 × 512, with pixel sizes of 0.43 × 0.43 μm and z-spacing of 0.75 or 2 μm.

## Measurement of fluorescence intensity of F-actin.
In double-labeling (anti-MYO7A antibody and rhodamine phalloidin) samples, fluorescence intensity of rhodamine phalloidin was measured as follows. First, MYO7A positive cells were automatically detected by the custom-made program. The space without MYO7A staining was simultaneously detected as a putative position of cell loss. Second, rhodamine phalloidin fluorescence intensity was measured using ImageJ software. OHCs surrounded by intact OHCs and nearby OHCs next to the cell loss positions were selected. Finally, the rhodamine phalloidin intensities between different positions of the organ of Corti were normalized by the intensity of OHCs surrounded by intact OHCs (*Figure 5A*).

## Measurement of tissue transparency and size change
Tissue transparency was measured according to the methods described by Hama et al (*Hama et al., 2015*). Light transmitted through 100 μm-thick mouse brain sections before and after clearing was captured with a CCD camera and quantitated. Transparency was normalized to the samples without clearing treatments. Increase in tissue size was determined by

measuring the area of brain sections before and after clearing. The extent of size increase was expressed as a ratio against the pre-cleared area (*Figure 1—figure supplement 1*).

## Imaging depth quantification

To assess the imaging depth of bony tissue, 20 µg of Alexa Fluor 488-conjugated anti-mouse VE cadherin antibody was administered into tail vein followed by tissue fixation. The metaphysis of the tubular bone was analyzed. Imaging depth was measured from the surface and the position at which specific fluorescence signal of the vasculature over the background could no longer be detected was taken as the limit of the imaging depth (*Figure 1—figure supplement 2*).

## Noise exposure

Mice were placed within steel wire cages (20 cm $\times$ 12 cm $\times$ 7 cm) in a ventilated sound exposure chamber. The acoustic stimulus was an octave band noise with a center of 4 kHz at 121 dB sound pressure level (SPL) for 4 hr. The sound-delivery speakers (HFD-261–8, TOA) were driven by a power amplifier (IP-600D, TOA) attached to a noise generator (AA-61B, RION) through a programmable filter (3624, NF Corporation). Sound levels were measured (UC-31 and UN14, RION) at multiple locations within the sound chamber and calibrated (NC-74, RION) to ensure uniformity and stability of the stimulus.

## Auditory Brainstem Response (ABR) tests

ABRs were recorded before and 7 days after noise exposure. Mice were anesthetized with ketamine hydrochloride (50 mg/kg, i.p.) and xylazine hydrochloride (10 mg/kg, i.p.). Before the ABR test, the tympanic membranes were confirmed to be normal. ABRs were evoked with tone-bursts (5 ms duration at 4, 8, 16, and 31.25 kHz) and measured by a recording system (Neuropack $\sum$ MEB2208, Nihon Kohden). Needle electrodes were placed subcutaneously at the vertex of the skull, under the ear exposed to noise, and under the opposite ear for ground. A speaker was placed 10 cm from the tragus of the stimulated ear. ABRs from 500 trials were averaged at each sound intensity. ABR thresholds were estimated by changing the intensity in 5 bB steps and finding the lowest sound intensity where reliable response peaks were detected

## Statistical analysis

C57BL/6J (PND 5: four samples, PND 30: 10 samples, PND 60: 14 samples, PND 120: nine samples, PND 360: four samples, and PND 60 with noise: 10 samples) mice and CBA/Ca (PND 60: five samples) mice were analyzed. Statistical analysis was performed by paired *t*-test (*Figure 2G*), Kruskal-Wallis test with Steel-Dwass multiple comparison test (*Figure 3B*), paired *t*-test followed by Bonferroni's correction (*Figure 3D*), Welch's *t*-test (*Figure 4A*), paired *t*-test with Bonferroni's correction and Welch's *t*-test with Bonferroni's correction (*Figure 5*), one-way ANOVA followed by Bonferroni's post hoc test (*Figure 1—figure supplement 1* and *Figure 2—figure supplement 1A*), or a two-tailed unpaired *t*-test (*Figure 1—figure supplement 2A* and *Figure 2—figure supplement 1B,C*). Data are presented as means ± standard errors of the mean (SEM) except standard deviation (SD) in *Figure 4E*. The p-values are as follows: *p < 0.05, **p < 0.01, ***p < 0.001.

## The auto cell-count and three-dimensional morphology analysis

The outline of the protocol, including stitching of multiple image stacks into a single stack, reconstruction of linearized image, automated detection of IHCs, and automated detection of OHCs, was described in the main text. Full description of the algorithm is in Appendix 2.

## Frameworks of machine learning models

The outline of the framework was described in the main text. Full description of the models is in Appendix 2.

## Validation of the automated OHC loss detection and counting

To evaluate the performance of automated OHC loss counting program, three skilled human operators (A, B, and C) initially detected empty spaces in the same set of the images (n = 4). A human consensus (HC) was established for the presence of empty spaces when more than two operators agreed. The match rates among human operators (A-B, B-C, and A-C) and between the program and HC were comparable (A-B: 91.6, B-C: 89.8, A-C: 89.7, Auto-HC: 93.1, *Table 3*). The numbers of lost cells in individual empty spaces were then estimated by both the human operators and the program using 161 empty spaces (*Figure 2G*). The differences between the human operators and the program were defined as absolute errors, and the errors were averaged (mean absolute error) and compared between the program and the human operator (Auto-M1, Auto-M2, and Auto-M3) or between the human operators (M1-M2, M2-M3, and M1-M3).

## Analysis of lost cell distribution in the organ of Corti

The longitudinal pattern of cell loss was analyzed among five segments ('a', 'b', 'c', 'd', and 'e') along the longitudinal axis (x-axis). The two segments from the basal end with their lengths of 800 μm ('a' and 'b'), two segments from the apical end with their lengths of 800 μm ('e' and 'd') were defined, and the middle remaining segment was defined as 'c' (*Figure 3B*). The radial axis (y-axis) was divided into 15 successive zones from the proximal to the distal, covering the width of three rows of OHCs (*Figure 3D*).

## Appendix 2

DOI: https://doi.org/10.7554/eLife.40946.023

# Supplemental materials and methods for data processing

MATLAB source code is available on GitHub (https://github.com/okabe-lab/cochlea-analyzer.git).

## Datasets for automated detection and analyses

For automated detection and analyses of hair cells, we examined 43 sets of images from 43 samples (PND 30: 10 samples, PND 60: 14 samples, PND 120: nine samples, noise exposure at PND 60: 10 samples). The samples immunostained with MYO7A were imaged with a two-photon microscope. To cover the entire longitudinal length of the organ of Corti, multiple image stacks with overlapping volumes should be obtained. Single image stack consists of 2D images with their pixel sizes of 512 × 512, with image numbers in the range of 36–459 along the z-axis. Large differences in the stack sizes were due to changes in the height of the fluorescent objects along the spiral of the cochlea. The image data set for a single cochlea sample contains 11–16 image stacks, which follow the spiral organization of the organ of Corti from an apical end toward the hook portion. There were 10–40% of overlap in length between adjacent image stacks. Physical longitudinal lengths of the organ of Corti were relatively uniform across samples (*Figure 2C* in the main text). The voxel sizes in x, y, and z directions were 0.99, 0.99, and 1.0 μm, respectively.

## Outline of automated cell detection and analyses

The scripts written for computational detection and analysis of hair cells consist of the following steps:

   Step 1: Stitching of multiple image stacks into a single stack
   Step 2: Reconstruction of linearized image
   Step 3; Automated detection of IHCs
   Step 4; Automated detection of OHCs

## Step 1: Stitching of multiple image stacks into a single stack

Multiple image stacks containing parts of the organ of Corti were assembled into a single image stack (*Rankov et al., 2005*). The cross-correlation method was used to image stitching. For rough estimation of the relative positions of two adjacent image stacks, we used recorded positions of the microscope stage and the overall distribution of fluorescent signals derived from the organ of Corti after appropriate image thresholding. Relative positions of two adjacent images were determined by searching best cross-correlation of images at the putative region of overlap (MATLAB 'normxcorr2').

   Overlapped images were blended with gradient mixing of intensities. To simplify the explanation, the following protocol is for two-dimensional blending. Blending of image stacks could be performed with similar principles by setting blending gradient in three dimensions. The mixing ratio of the two images was calculated using a sigmoid function as:

$$R_1 = \frac{1}{1 + e^{-10x}} \, ,$$

$$R_2 = 1 - R_2 \, ,$$

where $R_1$ and $R_2$ denote mixing ratios of the first and the second images, respectively. The value $x$ was set to reflect the relative distance of each pixel from the line running through the points with equal distance from two image centers (the center line). The normalized distance

was measured from the center line, where both $R_1$ and $R_2$ were 0.5. $R_1$ reaches ~1 as $x$ moves toward the center of the first image, while $R_2$ reaches ~0. Because the blending process ends at the pixel position within the image overlap that has the largest distance from the center line, value $x$ was normalized to be 1 at this largest distance (*Appendix 2—figure 1*).

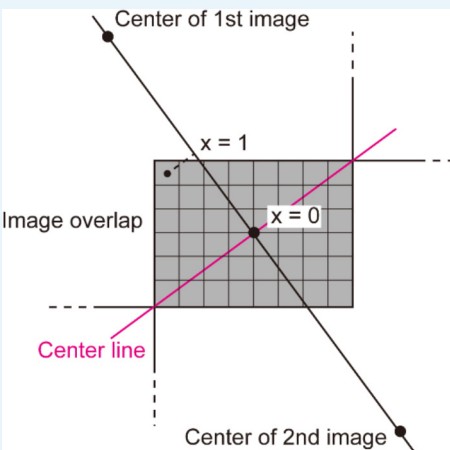

**Appendix 2—figure 1.** First the line passing through the centers of two images were generated, and the line passing through the center of the image overlap and perpendicular to the first line was created (the center line). Distance of each pixel to the center line was defined as $x$. The pixel that has the largest distance was selected and its $x$ value was normalized to be 1.
DOI: https://doi.org/10.7554/eLife.40946.025

## Step 2: Reconstruction of linearized images

After Image processing by a median filter (MATLAB 'medfilt3', neighborhood size: 3-by-3-by-3), local intensity peaks were automatically detected (MATLAB 'imregionalmax, 26-connected neighborhoods). Candidate anti-MYO7A-derived signals were selected from the local intensity peaks by eliminating those with their intensities below the threshold value α determined by the following formula.

$$\alpha = \mu + 5\sigma,$$

where $\mu$ and $\sigma$ denote the mean and the standard deviation of image intensities in the background. Otsu's method was used to determine the background thresholds (MATLAB 'multithresh', number of threshold values: 2, lower value was used) (Otsu, 1979). To further eliminate the intensity peaks derived from non-specific fluorescence, single-linkage clustering was performed (maximal distance of connection, 25 $\mu$m). This procedure was effective in identifying clustered intensity peaks corresponding to the organ of Corti as the largest assembly of linked intensity peaks.

The stretch of these intensity peaks corresponded to the entire row of hair cells. In the next step of the analysis, more precise determination of the structural parameters for the sensory epithelium is required. For this purpose, the spiral of the hair cell rows should be divided into segments of 200–300 µm in length. Because the sensory epithelium is curved in each original image stack and the spatial distribution of the intensity peaks roughly follows the shape of the sensory epithelium, principal component analysis [PCA (MATLAB 'pca')] was effective in extracting the direction and distance in both longitudinal and radial axes of the organ of Corti (*Appendix 2—figure 2*). Intensity peaks within a single image stack were divided into two equal halves along the first principal component axis, which were generally 200–300 µm in length. Both were projected onto PCA plane defined by the first and second principal component axes, and the best-fit arcs were calculated. The center and the radius of the arc were determined to minimize the mean square error (MATLAB 'fminsearch'). Repetition of this procedure for all the original image stacks gave rise to a collection of the sensory epithelium segments, that encompassed the entire spiral of the organ of Corti.

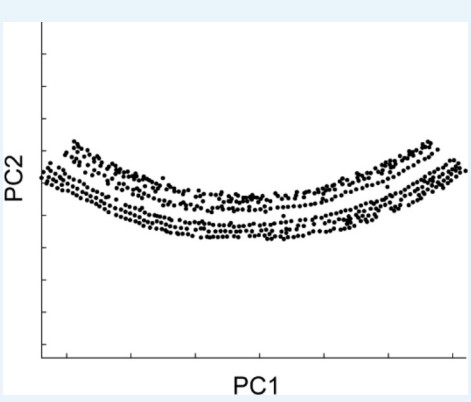

**Appendix 2—Figure 2.** Distribution of intensity peaks within the plane of the first and second principal components. The first principal component matched the longitudinal axis, while the second matched the radial axis.

DOI: https://doi.org/10.7554/eLife.40946.026

The multiple best-fit arcs were stitched into a continuous curve. The nearest-neighbor points of two adjacent arcs were detected (*Appendix 2—figure 3*). Starting from these nearest-neighbor points, both of the arcs were converted to the connection of points with regular intervals of 50 μm. For the calculation of a spline curve that fits two adjacent arcs, the points along each arc were divided into two groups at the position of the nearest-neighbor point and the groups of points that belong to the main portion of the sensory epithelium were selected and combined into a single group. The nearest-neighbor points themselves were excluded from the combined group. The combined points were interpolated by using the spline interpolation method (MATLAB 'interp1', method: 'spline') and principal normal vectors were calculated (hereinafter just called 'normal vectors').

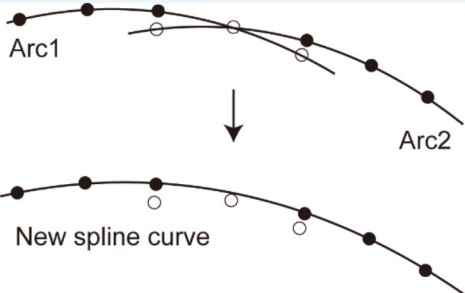

**Appendix 2—Figure 3.** The method of stitching two arcs. Among the dots on the two arcs, the open dots were removed and the closed dots were fitted with a spline curve.

DOI: https://doi.org/10.7554/eLife.40946.027

A voxel image containing the entire straightened rows of hair cells was reconstructed based on the stitched fit curve and the normal vectors. The horizontal center line (x axis) of the reconstructed image was set to match the fit curve. The y axis of the reconstructed image corresponds to the normal vector of the fit curve. The sizes of a voxel along the x, y, and z axes in the reconstructed image were all set to 1.0 μm. As each voxel in the linearized image does not show one-to-one correspondence to the original voxel image, interpolation was necessary to estimate the voxel intensity. In general, when a single voxel in the first image was projected to the second image, it was possible to define $2 \times 2 \times 2$ voxels in the second image enclosing the volume projected from the first image. Therefore, the intensity of a voxel in the linearized image was estimated by the trilinear interpolation method using the corresponding $2 \times 2 \times 2$ voxel in the original voxel image. More precise linearization was performed using a

fit curve of IHCs detected by the procedure described in the next section. The voxel values are set to zero when the corresponding voxels are outside the original images.

## Step 3; Automated detection of IHCs

Template matching was performed with a template image of an IHC (MATLAB 'normxcorr2', template image size: 11 × 11 pixels). This process puts a correlation coefficient to each element of a three dimensional matrix, which reflects the degree of matching to the template. By conducting two-dimensional peak detection on this matrix (MATLAB 'imregionalmax'), candidate voxels for IHC positions were selected. The connected-component labeling was then carried out on the binary matrix of candidate IHCs (26-connectivity based). These connected pixel groups (hereinafter called 'cell candidates', *Appendix 2—figure 4*) were used as the first approximation of IHC positions linked to other attributes, including correlation values and local intensity distributions. The cell candidates were further evaluated to eliminate false positives using two successive machine learning models (the overview of our method related to machine learning is described in 'Principles of auto-detection with machine learning' section below).

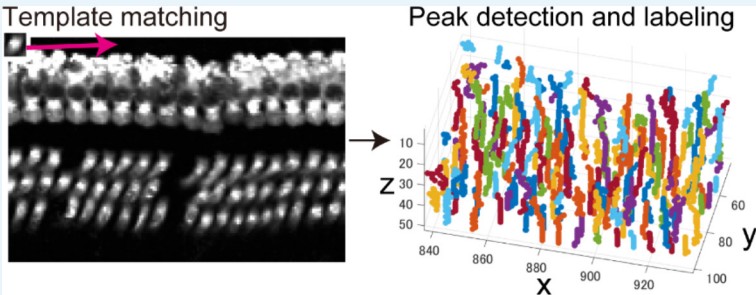

**Appendix 2—figure 4.** Extraction of cell candidates by template matching. Left; Template matching with the small template image (upper left) was performed for individual x-y images within the image stack. Right; Detection and labeling of correlation peaks distributed within three-dimensional matrix generated by calculation of cross-correlation. Cell candidates were assemblies of correlation peaks grouped by connected-component labeling.

DOI: https://doi.org/10.7554/eLife.40946.028

In the first step, the cell candidates were narrowed down to the number approximately twice larger than the expected number of IHCs. Specifically, the typical initial number of cell candidates was ~50,000 per one linearized image and they were narrowed down to ~ 1000 in this step. An ensemble learning method was employed to create the model for this selection (MATLAB 'fitensemble', method: 'GentleBoost') (*Friedman et al., 2000*). The predictor data was composed of the area, barycentric coordinates, correlation values, peak intensities, and those of nearby cell candidates. The model was trained to calculate posterior probability (prediction score) that the cell candidates corresponds to IHCs. The cell candidates with high prediction scores were selected for the next step.

In the second step, further selection of the cell candidates was performed by another machine learning model with expanded predictor data. The model was built based on an ensemble learning method (MATLAB 'fitensemble' function with 'Bag' method) (*Breiman, 2001*). The expanded predictor data was composed of the prediction score from the first step of the cell candidates and nearby cell candidates, and a small intensity image centered on the barycentric coordinates of the cell candidate.

Finally, the cell candidates were connected sequentially using physical constraints that the IHCs form a single row with roughly constant intervals of more than 6 μm along the x-axis. The obtained coordinates of estimated cells were used for fine readjustment of image linearization and the three dimensional structural analysis of the whole cochlea.

## Step 4; Automated detection of OHCs

The first several steps of cell detection protocol were similar to those for IHCs described above. In short, template matching was performed with a small template image of OHCs (image size: 7 × 7 pixels) to create the matrix of correlation coefficients. The connected-component labeling was carried out on the binary matrix for the positions of the correlation peaks. The groups of peaks were taken as the cell candidates and narrowed down through two steps of prediction by two machine learning models. Ensemble learning methods were employed to build these models with similar predictors as described above.

For accuracy improvement, several steps were added after the second prediction step. As OHCs are typically distributed in three rows, the selected cell candidates were classified into three rows by a multiclass classification model (*Appendix 2—figure 5*). The model was built based on the convolutional neural network method (Neural Network Toolbox of MATLAB) (*LeCun et al., 1989*). The input of the model was a small image centered on the estimated cells. To reduce the chance of missing OHCs, regularity of distances between adjacent cell candidates in row was investigated. If the distance between two adjacent cell candidates in the same row exceeds 1.5 times of average distance, presence of additional cells in the gap was judged by the fourth model.

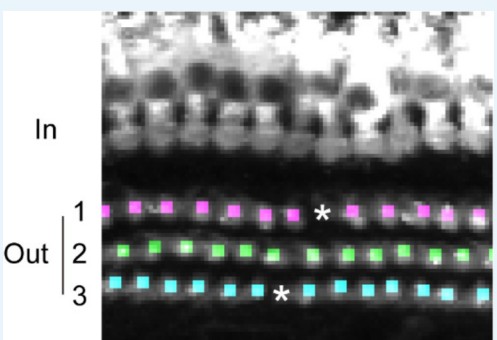

**Appendix 2—figure 5.** Functions of the third and fourth machine learning models. Cell candidates detected by the first and the second machine learning models were further categorized into three rows by the third model (cells marked by dots with different colors correspond to the three rows.). The spaces between detected OHCs (asterisks) were detected and evaluated the possible presence of OHCs escaped in the previous detection processes.
DOI: https://doi.org/10.7554/eLife.40946.029

To evaluate the missing cells in the space, the forth model was used. This model is also built based on the convolutional neural network method and uses small images as an input. Input small images for machine learning were sampled by placing small rectangular areas as region of interest (ROI) with equal distance from each other within the gap between preexisting cell candidates. The number of ROI per space was set to be equal to the integer portion of the quotient with the gap length as a numerator and the average distance of adjacent cells as a denominator. In the case that the model predicted the existence of a cell at the space, the nearest peak of correlation coefficient by the first template matching was added to the list of estimated cells. The obtained coordinates of estimated cells were used for the analysis of cell loss.

## Remarks on principles and realization of data processing

## Comparison of detection efficiency with standard image processing method

Performance of our automated cell detection system (Step 3 and 4) was compared with a classical image processing method, a three-dimensional watershed method, with optimized

parameters, such as threshold values for intensity and volume. The accuracy (F score) was ~ 70% on average, corresponding to ~ 600 false positives and ~ 600 false negatives in a linearized image containing 2500 hair cells (*Appendix 2—figure 6*, *Table 2*). This performance is not adequate for practical use. The details of the watershed technique are described below ('Auto-detection with three-dimensional watershed algorithm' section).

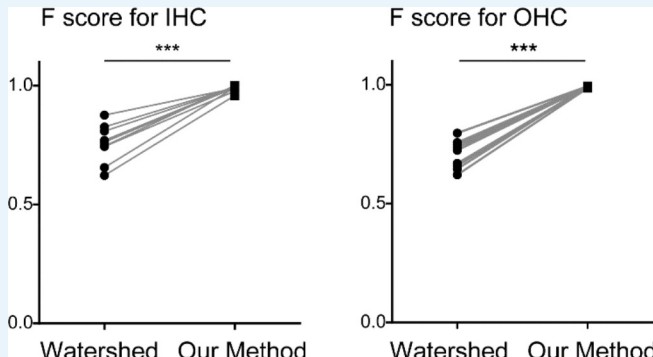

**Appendix 2—figure 6.** Comparison of detection efficiencies between a standard image processing method and our method (Paired t-test, both p < 0.0001, n = 10 linearized whole cochlear images).

DOI: https://doi.org/10.7554/eLife.40946.030

The poor performance of the standard watershed method may be derived from several reasons (*Appendix 2—figure 7*). First, hair cells were often counted more than once, because of multiple intensity peaks within a single hair cell. The watershed method typically selects seeds from peaks of intensity in the image. Reduction of peaks can be done by image smoothing with a Gaussian filter before the watershed method. However, optimization of Gaussian filter size was difficult, as hair cell shapes are highly heterogenous at different positions of the cochlea. The second reason was false detection of nonspecific signals of anti-MYO7A antibody and background noise. This error can be reduced by a volume, intensity or morphological filtering, but optimization of filter settings was again difficult for heterogeneous hair cells along the longitudinal axis of the sensory epithelium.

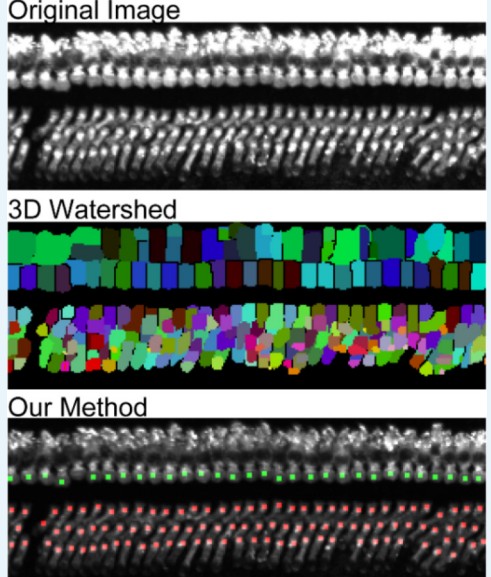

**Appendix 2—figure 7.** Comparison of 3D watershed and our machine learning based method. There are many duplicate count and false detection with 3D watershed method.

DOI: https://doi.org/10.7554/eLife.40946.031

To achieve practical accuracy for cell detection in a large data set of whole cochlear image, we designed the method based on multiple pattern recognition techniques. We overcome the first problem, duplicated cell counting, with template matching technique (*Appendix 2—figure 4*). The second problem, noise filtering, was addressed by machine learning models as described in the next section. To further improve the accuracy of detection, we also added a step of recovering false negatives based also on machine learning that detected three-row arrangement of outer hair cells.

## Principles of auto-detection with machine learning

Our machine learning based auto-detection system utilizes two principles. The first principle is sorting hair cells from background noise. The models were trained with labeled dataset containing multiple feature values of cell candidates, such as volume, signal intensity, similarity with template image, coordinates, and relative position from neighboring candidates (*Table 1*). As morphology of hair cells and imaging conditions vary from place to place within the image, it was almost impossible to find the single or a few feature values that could efficiently distinguish cells from background noise. The approach with machine learning models, however, dramatically improved the accuracy of cell detection (*Appendix 2—figure 6*). Incorporation of excessive feature values may introduce a risk of overfitting. It is important to select machine learning algorithms resistant to the overfitting problem. After evaluation of available supervised learning algorithms in MATLAB ('Statistics and Machine Learning Toolbox' in MATLAB R2017b; Decision trees, Discriminant analysis, Nearest neighbors, Naïve Bayes, Support vector machines, and Classification ensembles), we chose classification ensembles (Gentle AdaBoost and Random forest algorithms) because of their high generalization abilities and processing speeds. Training dataset from ten cochlear samples and test dataset from other four samples were used.

The second principle is recovery of false negatives based on auto-recognition of three-row arrangement of outer hair cells (*Appendix 2—figure 5*). Although the detection based on the first principle shows high specificity of hair cell detection, it sometimes misses cells with atypical morphology or features. The first step of the false negative recovery is grouping of each cell candidate into one of three rows. Then, spatial gaps are detected by checking irregularity of intervals of cell candidates within each row. Finally, the gaps were evaluated whether there were overlooking cells or not by another model. This step of false negative recovery was realized by a convolutional neural network algorithm. The advantage is that it only requires an image as an input without any additional feature extraction. We expected the complementary effect by combining the first and second principles. Indeed, the recovery rate of the detection was further improved by adding the second process (*Table 2*, Paired *t*-test, $p < 0.005$, $t = 3.94$, $df = 9$).

## Auto-detection with three-dimensional watershed algorithm

We tested another auto-detection method for linearized image implemented in 'Classic Watershed' plugin for Fiji (*Soille and Vincent, 1990*). Several settings should be configured for this approach. First, a threshold value for voxel intensity should be provided. With lower threshold, more cells will be detected, but false positives will also increase. A threshold value for volume of the segmented region is also required for reducing false positives. In addition, inner and outer hair cells were distinguished based on their center coordinates along the radial axis in the linearized image. We optimized these parameters by grid search on the test samples for maximizing the detection efficiency (F score). Fiji macro and MATLAB source code of this method is available on GitHub (*Iida, 2018b*; copy archived at https://github.com/elifesciences-publications/Watershed).

## Training of machine learning models

The models were trained with the dataset obtained from ten cochleae with manual labeling (PND60: two samples, ACL: five samples, NCL: three samples). The numbers of observations of the training datasets are shown in *Table 4*. Details of the predictors of datasets are shown in *Table 1*. The datasets for the models IHC1 and OHC1 include all the peak groups obtained from the ten linearized images with template matching (*Appendix 2—figure 4*). These models are based on a GentleBoost algorithm. To deal with the imbalance between positives and negatives, we set the cost matrix when training IHC1 and OHC1 (100 times and 50 times higher costs for false negatives than false positives, respectively). The hyperparameter was optimized by five-fold cross validation with a Bayesian optimization method on the training dataset (number of the trees, learn rate; 'OptimizeHyperparameters' option for 'fitcensemble' function of MATLAB).

A subgroup of cell candidates was selected for further processing with the models ICH2 and OHC2, based on their higher posterior probability computed by IHC1 and OHC1. The fraction of this subgroup over the total sample population was fixed. The models ICH2 and OHC2 were based on a Random Forest algorithm. We set the cost matrix to deal with the imbalance of the dataset when training IHC2 and OHC2 (2 times and 0.4 times costs for false negatives, respectively). The hyperparameter was optimized as with IHC1 and OHC1.

The training dataset for the model OHC3 includes small images centered on each outer hair cell extracted from ten linearized images. This model is based on a CNN algorithm. We used a shallow network with typical settings for training. Details of the layer configuration (filter size and number of convolutional layer) were optimized by five-fold cross validation with a Bayesian optimization method on the training dataset. We trained the network with stochastic gradient descent with momentum algorithm, with the maximum number of epochs of 15, and the initial learning rate of 0.001 (*Murphy, 2012*). We used the default settings of 'trainingOptions' function of MATLAB for the other training options.

The dataset for the model OHC4 includes small images of gaps between cell candidates within each row of outer hair cells (*Appendix 2—figure 5*). The model was based on a shallow CNN with typical settings for training. We optimized the details of the layer configuration of the model and trained it in the same way as OHC3.

## Evaluation of machine learning models

The models were tested by ten cochleae that were not used in the training (PND30: two sample, PND60: three sample, ACL: two sample, NCL: three sample). The way of making the test datasets was the same as the training dataset. The numbers of the test dataset of each model are shown in *Table 4*. The indices of the performance were obtained as follows:

$$Recall = \frac{TP}{TP+FP} \, ,$$

$$Precision = \frac{TP}{TP+FN} \, ,$$

$$F\ score = \frac{2 \times Precision \times Recall}{Precision+Recall} \, ,$$

where TP, FP and FN mean True Positive, False Positive and False Negative, respectively. For multiclass classification, the micro-averages of recall and precision were calculated (*Sokolova and Lapalme, 2009*).

### Details of procedures for the analysis of cochlear structure and hair cell position

## Analysis of three-dimensional structure of cochlea

Spiral structures of the organ of Corti were analyzed using three dimensional spatial distribution of IHCs. The distribution of IHCs was used because they form a row with little cell loss. The coordinates of IHCs in the linearized image were transformed inversely into the original three dimensional coordinates based on the fit curve and the normal vectors which were also used in the forward linearization.

A cylindrical coordinate system of each sample was set for registration and comparison between samples. In this system, the position of the $i$ th IHC along the organ of Corti is expressed as the combination of the longitudinal coordinate ($p_i$), the azimuth ($\varphi_i$), and the radial coordinate ($\rho_i$). Please note that here we refer the direction of the modiolus from the apex to the base as the longitudinal axis. In the previous description of the protocol, we refer the direction along the long axis of the sensory epithelium as "longitudinal".

The strategy of the initial search for the longitudinal axis was to find the arguments of the minimum for the function that gives smaller values when a line (longitudinal axis) is set to provide the spiral fitting better to the distribution of IHCs. The longitudinal axis $l$ of the coordinate system was chosen as follows:

$$l = arg\min_{l_i \in L} f(l_i),$$

where $L = \{l_1, l_2, \ldots\}$ is a set of lines in three-dimensional Euclidean space and $f(l_i)$ is the objective function as:

$$f(l_i) = \min_{a,b \in R} \sum_{j=1}^{n} \left( \rho_j^i - \left( a + b\varphi_j^i \right) \right)^2,$$

where IHCs in a sample were numbered in the order of distance from the apical end along the organ of Corti, and $n$ denotes the total number of IHCs. Values $\rho_j^i$ and $\varphi_j^i$ denote the radial coordinate and the azimuth respectively, for the $j$-th IHC in a coordinate system with a given longitudinal axis $l_i$. The azimuth $\varphi_1^i$ for the first IHC located at the apical end was set to 0 and the others were set to satisfy the condition:

$$\varphi_{j-1}^i < \varphi_j^i < \varphi_{j-1}^i + 2\pi.$$

The objective function $f(l_i)$ means the goodness of fit (the minimum sum of squared errors) of IHCs' locations to a curve defined by the equation:

$$\rho = a + b\varphi,$$

where $\rho$ is the radial coordinate, $\varphi$ is the azimuth (**Appendix 2—figure 8** and **Appendix 2—figure 9**). Parameter optimization was performed by using the 'fminseach' function of MATLAB. The longitudinal axis $l$ that provides the arguments of the minimum was considered as the modiolus in this protocol. The distance and the cell number from the basal end of the IHC row along the spiral connecting IHCs was measured ('Distance from Base' and 'Cell Number' in **Figure 2** of the main text).

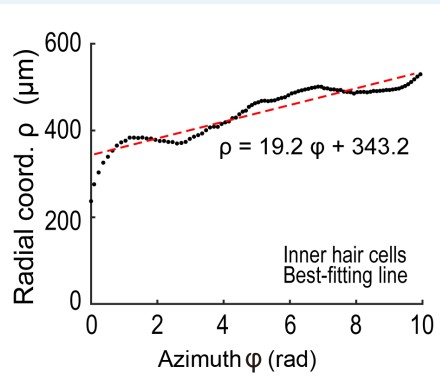

**Appendix 2—figure 8.** An example of fitting the increasing distance between IHCs and the modiolus to a smooth spiral. The line with the minimum sum of squared errors was chosen to be the longitudinal axis of the cylindrical coordinate system.
DOI: https://doi.org/10.7554/eLife.40946.032

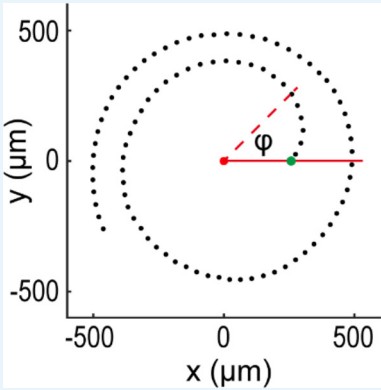

**Appendix 2—figure 9.** An example of inner hair cell locations (black dots) viewed from the axial direction. The angle ($\varphi$) was measured from the line (red line) connecting the center of the spiral and the inner hair cell located at the end of the apex (green dot).
DOI: https://doi.org/10.7554/eLife.40946.033

## Registration of samples based on 3D structure

In the previous section, the longitudinal axis (modiolus axis) was set for each sample and utilized for the presentation of the IHC spiral in the cylindrical coordinate system. Registration of multiple samples were performed using these parameters. To reduce sampling bias in the alignment process of multiple samples, we followed the strategy of first creating a generic template of the IHC spiral by averaging all the available samples. After this step, alignment of individual samples to the generic template was performed. Nevertheless, in both cases we utilized the common protocol for alignment. To simplify the following explanation, we refer two spirals to be aligned as spirals A and B (*Appendix 2—figure 10*).

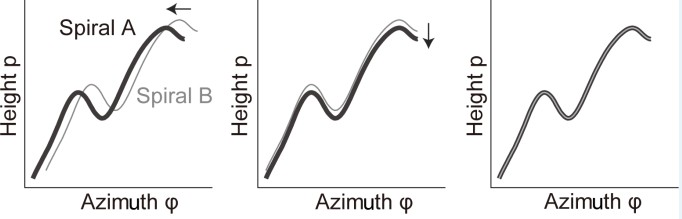

**Appendix 2—figure 10.** Evaluation of the extent of correlation between two curves in the plane

of $\varphi$-$p$. Within this plane, the positions of spiral A and B were aligned. First, the shift in $\varphi$ was adjusted (middle). Subsequently, the shift in $p$ was adjusted (right).

DOI: https://doi.org/10.7554/eLife.40946.034

We first performed alignment between spirals A and B by rotating objects around the longitudinal axis. This can be mathematically achieved by shifting the value $\varphi$ of one spiral against the other. To estimate the amount of shift in $\varphi$, a two-dimensional plot of $p$ against $\varphi$ was created and overlaid for spirals A and B. By evaluating the extent of correlation between two curves in the plane of $\varphi$-$p$, the optimized shift in $\varphi$ can be estimated. To calculate the extent of correlation, two curves were converted to series of points with their angular intervals of 0.1 rad and their cross-correlation was calculated (MATLAB "normxcorr2"). The shift in $\varphi$ that gives the highest cross-correlation was selected and the rotational shift between two curves A and B were corrected.

The last step of alignment between spiral A and B was translation along the longitudinal axis. To this end, the spiral A was fixed and the origin of the second spiral B was shifted along the longitudinal axis. Comparison of the two spirals was based on the search for the minimum sum of the squared distances along the longitudinal axis between points from two spirals. These points were selected within the region of overlapping azimuth for the two spirals. Two corresponding points on different spirals share the same $\varphi$. The origin of the second spiral was determined by finding the arguments of the minimum for the function of squared distance between corresponding points for two spirals projected to the longitudinal axis as follows:

$$O = arg\,min_{p_i \in P} \sum_{\varphi \in \Phi} (z_0(\varphi) - z_i(\varphi))^2,$$

where $P = \{p_1, p_2, \ldots\}$ is a set of positions on the longitudinal axis $l$, and $\Phi$ is a set of azimuth $\varphi$ shared by the points of comparison between spirals A and B. The function $z_o(\varphi)$ denotes an axial coordinate of the point on the spiral A. The function $z_i(\varphi)$ denotes an axial coordinate of the point on the spiral B when the origin of the spiral was set at the position $p_i$. The values 'Distance from Base ($\mu$m)' (**Figure 2** and **Figure 3** in the main text) and 'Cell Number' (**Figure 2** in the main text) were also plotted after alignment at the midpoint of the spiral segments.

## Analysis of spatial distribution of OHCs

The outline of the procedure was described in the main text. Loss of OHCs results in the formation of empty spaces in the epithelium. Conventionally, the lost cell number was estimated by the size of the empty space. This method is not reliable when the empty space is large, the rows of OHCs are difficult to define, or cell-to-cell distances vary in different epithelial positions (**Appendix 2—figure 11**).

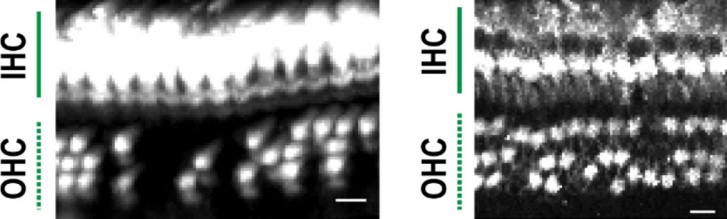

**Appendix 2—figure 11.** Representative examples of hair cell images where manual estimation of cell loss was difficult. The number of lost cells is difficult to estimate when the size of cell-negative area increases in the basal turn (left). Disorganized rows of OHCs were frequently observed in the apical turn (right). Identification of lost cell positions is difficult when cell density is not high enough to estimate the rows (left) or cells are not aligned as horizontal rows (right). Scale bars, 10 μm.

DOI: https://doi.org/10.7554/eLife.40946.035

In this study, we evaluated the space where cells were lost without assuming OHC rows. First, the x and y coordinates of the centers of detected cells (hereinafter called 'cell centers') were measured in the linearized image and created a scatter plot. Second, we obtained parameters necessary for radial alignment (along y-axis) of cell centers. For this purpose, we searched the longitudinal positions (along x-axis) along the organ of Corti, where a zone with the width of 8 μm (a gray zone in *Appendix 2—figure 12*) contained more than two cell centers (three black points within the gray zone in *Appendix 2—figure 12*). When the zone was found, two values, the averaged x coordinates (red circle in *Appendix 2—figure 12*) and the largest and smallest y coordinates (red line in *Appendix 2—figure 12*) of cell centers, were calculated for this zone. The former value was used as a reference of the center position of the epithelium and the latter value as a reference of its width ($Wy$). The vertical widths (subtraction of the largest and smallest y coordinates) were interpolated linearly along the x-axis and used for normalizing the y coordinates of cell centers. Third, we obtained parameters necessary for longitudinal alignment (along x-axis) of cell centers. For this purpose, we searched all the cell centers and created rectangles with their edge lengths of 24 μm along the x-axis and 8 μm along the y-axis with their centers aligned with the cell centers (a red point and a gray rectangle in *Appendix 2—figure 13*). Distances between the original cell center and the center of the nearest cell within the rectangle along x-axis were calculated ($Wx$). The averages were calculated with 100 μm intervals along the x-axis and interpolated linearly. Fourth, the coordinates of the cell centers were normalized using averaged $Wx$ and $Wy$ values to equalize the horizontal and vertical distances and to keep the distances uniform along the organ of Corti (*Figure 2F* in the main text).

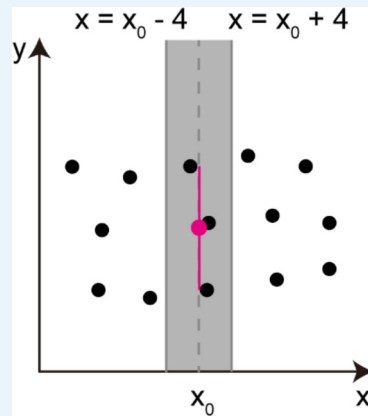

**Appendix 2—figure 12.** Procedures of obtaining parameters necessary for radial alignment (along y-axis) of cell centers. Calculation of an averaged y position of the cell group (a red circle) and a vertical spread of the cell group (a red vertical line). These two parameters were calculated in the area (colored in gray) containing more than two 'cell centers'. Black dots indicate the positions of 'cell centers', and the variable $x_0$ indicates the x-coordinate of the averaged cell center within the gray area.

DOI: https://doi.org/10.7554/eLife.40946.036

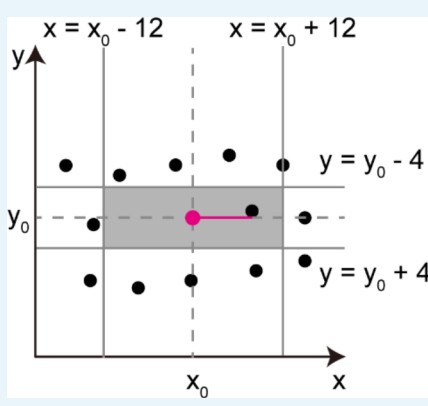

**Appendix 2—figure 13.** Procedures of obtaining parameters necessary for longitudinal alignment (along x-axis) of cell centers. Calculation of the horizontal distance between adjacent cells (red horizontal line). The nearest cell in the rectangular area (colored in gray) was selected for the calculation. The variables $x_0$ and $y_0$ are the coordinates of the parental cell center (red dot).

DOI: https://doi.org/10.7554/eLife.40946.037

A binary image of the normalized epithelium was created based on the equalized coordinates of cell centers (*Figure 2G* in the main text). The coordinates projected onto an image were adjusted to have the average distances between neighbors in x and y as five pixels. The horizontal center line of the image was set to be on the line y = 0. The height of image was set to 15 pixels and the width was adjusted to the range of x coordinates. Then squares of 5 × 5 pixels centered on each cell point were drawn on the image. Small holes were removed by a morphological closing operation. The empty spaces in the image were considered to be the putative cell loss sites.

The estimated amount of cell loss in the entire organ of Corti or in specific areas was shown as either the number of pixels in the empty spaces (*Figure 3D* and *Figure 3—figure supplement 1B*) or the cell number obtained by the formula $n = \frac{v}{25}$, where $n$ is the estimated number of cell loss and $v$ is the area of each connected region. The estimated number of cell loss was rounded off to the nearest integer (*Figure 3A,B* and *Figure 3—figure supplement 1A,C*). For the simulation analysis of cell loss models, the same formula was used for the estimated number of cell loss (*Figure 4* and *Figure 4—figure supplement 1*).

## In case of transferring to other programming language

The MATLAB scripts can be viewed on the website of GitHub without installation of MATLAB environment. The scripts have modular architecture, and the inputs and outputs of each module are annotated in the scripts. It would be relatively easy to transfer to the language which can preserve the architecture, such as C/C ++ or Python, by transferring the modules one by one. As there are several built-in functions specific to MATLAB (ex. 'findpeaks'), however, alternative means would be needed in some cases. Please refer to the website of MathWorks for the syntax and built-in functions of MATLAB in such cases (https://www. mathworks.com). In our scripts, the names of modules we made are written in upper camel case. The variables are written in lower camel case, and the constants in all capital letters. The names of built-in functions are written in lower case. Regarding how to train machine learning models, please refer to the 'Training of machine learning models' section above. Feel free to contact the corresponding author in case you have questions regarding the contents of the code. We would consider the transfer to other programming languages as needed.

