## [Decision Letter]

Thank you for submitting your article "Cellular cartography of the organ of Corti based on optical tissue clearing and machine learning" for consideration by *eLife*. Your article has been reviewed by three peer reviewers, and the evaluation has been overseen by a Reviewing Editor and Andrew King as the Senior Editor. The following individual involved in review of your submission has agreed to reveal his identity: Stefan Heller (Reviewer #1).

The reviewers have discussed the reviews with one another and agree that the manuscript requires minor revisions, which should be relatively quick to address. The Reviewing Editor has drafted this decision to help you prepare a revised submission.

Summary:

In this manuscript, Urata and colleagues describe a technical solution to minimize manual processing of cochlear samples from mice for quantitative analysis of hair cell numbers and location. The concept builds on a modified Sca/eS tissue clearing method, combined with immunolabeling of hair cells, 2-photon microscopy, and an elegant way of stitching image stacks and assembling a 3-dimensional image. The core of the automated analysis method is a series of MATLAB scripts that perform the analysis.

Essential revisions:

1) Please give more information about the methodology for the machine learning model and include a comparison with other methods. Transfer to other scripts is not essential, but please comment on whether such a transfer would be an option and how this might be done by someone interested in doing a similar analysis.

The full reviews are appended below.

Reviewer #1:

The MATLAB scripts appear well-documented and utilize common toolboxes. Unfortunately, this reviewer currently does not own a MATLAB license and therefore was not able to run the scripts to test for general applicability.

This brings me to my main critique. The beauty of this work is that it allows the researcher to process a large number of cochlear samples in parallel. With acquisition times of about 4h per sample using a 2-photon microscope and a 25x (NA = 1.1) objective and 30 min for an average analysis workflow, this method is indeed a major advancement for laboratories interested in an efficient and throughput-oriented method for quantitative assessment of hair cells and hair cell loss in the cochlea.

Nevertheless, the requirement for an expensive software package diminishes enthusiasm. What would it take to transfer the scripts to a widely available and free-for-all environment such as Python, Java, or R? Even C/C++ or a combination of languages comes to mind.

The overall approach is quite clever and this reviewer is enthusiastic about the work. Besides the automated data analysis, there is another very interesting hidden gem in this manuscript that this is related to Figure 4, and to the principles of observed clustering of lost hair cells and the two-component model. The use of a combination and dynamically changing (with age or noise challenge) model that takes into account local neighborhoods and position is truly a creative way to approach the analysis of such an observation. In this respect, I consider the results presented in this section of the paper and summarized in the last paragraph of the subsection “Model-based analysis of hair cell loss”, as a quite important and relevant finding.

A second critique concerns the core method used for data analysis. Machine learning is such a buzzword, but it is not a simple method that is replicable for the common reader. Exactly what kind of principle was utilized? Please provide accurate references already in the text presented in the second paragraph of the subsection “Machine learning–based automated detection of sensory hair cells”, and explain in more palatable fashion the kind of neural network approach that was used. Appendix 3 (Step 4) has some of this information, but the description there is difficult to follow and perhaps should be illustrated with some kind of drawing of the process. Appendix 2—figure 5 is a good start as it shows the sample that is investigated directly, but it does not communicate the process. Since this is an important component of the core method that is presented, one would expect a more detailed and more approachable description of the procedure.

Reviewer #2:

This manuscript by Urata et al. reports an application of a tissue clearing method for whole organ of Corti imaging and a following quantitative analysis. This reviewer thinks that the work is very important for the field, because a limited method had been applicable for the observation of the organ. The quantification and modeling analysis on the mechanism of hair cell loss due to aging and noise will also give an impact to the field. This reviewer thus appreciate the basic concept of the study, while I found many points which should be corrected and improved in future manuscript.

1) Please provide a simple and easily understandable figure indicating the step-by-step procedures of data processing according to the information in Materials and methods and Appendix part (e.g., Steps in Appendix 3) by modifying Figure 2. In the current manuscript, it was very difficult to follow and find relations of these steps in Figure 2 and Materials and methods/Appendix. Also, it seems problematic that there is no indication of procedure in Figure 2B (e.g., words such as "raw data" "stitching" "linearize" should be indicated in the panel).

2) The panels in Figure 3A and Figure 3—figure supplement 1A and the graph in panel B seem the same. Reuse of the same figure panel or data should be made explicit in figure legend. Is there any reason why the Figure 3—figure supplement 1A is elongated?

Reviewer #3:

This manuscript describes a method to image and quantitatively characterize the spatial arrangement of sensory hair cells in the organ of Corti. A sorbitol-based optical clearing method was developed to optimize tissue clearing and immunolabeling to improve tissue transparency and antibody accessibility. Two-photon microscopy was used to image the organ of Corti in 3D. The spiral sensory epithelium was then linearized and the positions of the inner and outer hair cells (IHCs and OHCs) were automatically located using a machine learning based algorithm. Using the new sample preparation and imaging analysis method, the authors analyzed age-related and noise-induced cell loss in young, adult, and noise-exposed mice.

One novel aspect of the paper is the modification of the Sca/eS optical clearing protocol, which first decalcifies the bones and then uses a guanidine-based solution to increase transparency of the sample with minimal tissue expansion and GFP fluorescence quenching. The modified Sca/eS method was compared to established methods, including 3DISCO, iDISCO, CLARITY, CUBIC and Sca/eS, and MYO7A and F-actin labeled hair cells were most clearly identified in samples treated with the modified Sca/eS protocol.

The authors then fit a spiral curve based on the 3D image information and linearized the entire sensory epithelium of the organ of Corti for further image analysis. This approach is quite impressive to me, although I have to admit that I am not an expert in this field and am not sure whether this approach (e.g. linearization of the entire sensory epithelium) is novel to the auditory research field.

Localization and classification of IHCs and OHCs using a machine learning method are nice. But very little details were provided to describe the convolution neural network model and how the training and testing were performed. This needs to be further clarified and fleshed out in the revised manuscript. It does not look like that the authors have improved the structure of the machine learning model. The number of samples used for training and testing the machine learning model is limited and can be improved.

Since the two-photon fluorescence images have very high contrast, a standard imaging processing method, such as 3D watershed, may just do the job with sufficient accuracy. It would be helpful to include a direct comparison of the machine learning model and the watershed algorithm. The reviewer also felt that the authors may consider downplaying the role of machine learning in the title and throughout the manuscript as traditional methods may achieve similar performance.

Based on the localization analysis of the IHCs and OHCs, age-related and noise-induced cell loss was quantified and compared between different mice models. I will not comment on the biology of the auditory system since this is beyond my expertise.

Overall, I commend the authors for providing a very careful analysis of the experiment results and assembled a comprehensive manuscript. Addressing the above comments may help to further strengthen the manuscript.

---

## [Author Response]

Essential revisions:1) Please give more information about the methodology for the machine learning model and include a comparison with other methods.

As suggested, we modified the main text to include brief description of the machine learning model (subsection “Machine learning–based automated detection of sensory hair cells”, second paragraph). In this description, we incorporated comparison with a different technique (3D watershed algorithm), which showed much lower performance for hair cell detection. For the details of the machine learning model, we inserted the description in the corresponding section (Materials and methods). In brief, we made six machine learning models in this research, with two for IHC detection (IHC1,2), four for OHC detection (OHC 1,2,3,4). Initial two IHC machine learning models (IHC1,2) are similar to two OHC models (OHC1, 2). IHC 1,2 and OHC1,2 are decision tree ensemble methods (GentleBoost and Random Forest) for discrimination of true signals and noise. OHC3, 4 are convolutional neural network models with small images for matching to templates.

Transfer to other scripts is not essential, but please comment on whether such a transfer would be an option and how this might be done by someone interested in doing a similar analysis.

Thank you for suggesting that transfer to other scripts is not essential for revision. We think the transfer will be relatively easy if the script could maintain the modular architecture of our code, and had built-in functions similar to MATLAB. We inserted several comments for the transfer in the corresponding section (Materials and methods).

The full reviews are appended below.Reviewer #1:The MATLAB scripts appear well-documented and utilize common toolboxes. Unfortunately, this reviewer currently does not own a MATLAB license and therefore was not able to run the scripts to test for general applicability.This brings me to my main critique. The beauty of this work is that it allows the researcher to process a large number of cochlear samples in parallel. With acquisition times of about 4h per sample using a 2-photon microscope and a 25x (NA = 1.1) objective and 30 min for an average analysis workflow, this method is indeed a major advancement for laboratories interested in an efficient and throughput-oriented method for quantitative assessment of hair cells and hair cell loss in the cochlea.Nevertheless, the requirement for an expensive software package diminishes enthusiasm. What would it take to transfer the scripts to a widely available and free-for-all environment such as Python, Java, or R? Even C/C++ or a combination of languages comes to mind.

As the Editor did not request transfer the scripts to other program languages, we maintained the current framework.

The overall approach is quite clever and this reviewer is enthusiastic about the work. Besides the automated data analysis, there is another very interesting hidden gem in this manuscript that this is related to Figure 4, and to the principles of observed clustering of lost hair cells and the two-component model. The use of a combination and dynamically changing (with age or noise challenge) model that takes into account local neighborhoods and position is truly a creative way to approach the analysis of such an observation. In this respect, I consider the results presented in this section of the paper and summarized in the last paragraph of the subsection “Model-based analysis of hair cell loss”, as a quite important and relevant finding.A second critique concerns the core method used for data analysis. Machine learning is such a buzzword, but it is not a simple method that is replicable for the common reader. Exactly what kind of principle was utilized? Please provide accurate references already in the text presented in the second paragraph of the subsection “Machine learning–based automated detection of sensory hair cells”, and explain in more palatable fashion the kind of neural network approach that was used. Appendix 3 (Step 4) has some of this information, but the description there is difficult to follow and perhaps should be illustrated with some kind of drawing of the process. Appendix 2—figure 5 is a good start as it shows the sample that is investigated directly, but it does not communicate the process. Since this is an important component of the core method that is presented, one would expect a more detailed and more approachable description of the procedure.

We added a description about principles of auto-detection in the corresponding section (Materials and methods). We utilized two principles for the auto-detection. One is a distinction of cells from background noise based on local feature values of each candidate point. Another is a process for the recovery of false negatives based on a recognition of the distributional pattern of hair cells.

Reviewer #2:This manuscript by Urata et al. reports an application of a tissue clearing method for whole organ of Corti imaging and a following quantitative analysis. This reviewer thinks that the work is very important for the field, because a limited method had been applicable for the observation of the organ. The quantification and modeling analysis on the mechanism of hair cell loss due to aging and noise will also give an impact to the field. This reviewer thus appreciate the basic concept of the study, while I found many points which should be corrected and improved in future manuscript.1) Please provide a simple and easily understandable figure indicating the step-by-step procedures of data processing according to the information in Materials and methods and Appendix part (e.g., Steps in Appendix 3) by modifying Figure 2. In the current manuscript, it was very difficult to follow and find relations of these steps in Figure 2 and Materials and methods/Appendix. Also, it seems problematic that there is no indication of procedure in Figure 2B (e.g., words such as "raw data" "stitching" "linearize" should be indicated in the panel).

As suggested, we modified Figure 2B for better understanding and added words for explanation of the procedure (“raw data”, “stitching”,”linearize”,”IHC counting”,”OHC & cell loss counting”) in the panel.

2) The panels in Figure 3A and Figure 3—figure supplement 1A and the graph in panel B seem the same. Reuse of the same figure panel or data should be made explicit in figure legend. Is there any reason why the Figure 3—figure supplement 1A is elongated?

As pointed out, the 2D color plot of cell loss in Figure 3A and the 2D color plot in Figure 3—figure supplement 1A are the same data. The reason we reuse the same data is that in supplement we provided the ID of each cochlear sample in the left edge of the color plot, in order to make explicit about the data identity. We agree that duplication of the same 2D color plot with different aspect ratio may be confusing to the readers and decided to delete the plot in the supplement. The IDs of the samples are now added to the second 2D color plot (along the proximal-distal direction).

Along the same lines, we decided to remove the duplicated bar graph attached to the 2D color plot in Figure 3—figure supplement 1B. We initially thought that this bar graph may help the readers to note differences in the total number of lost hair cells, but this can be done by comparing the panel A in Figure 3 and the (previous) panel B in the supplement.

Accordingly, panel alphabets were changed as follows.

“Figure 3—figure supplement 1A” => removed

“Figure 3—figure supplement 1B” => “Figure 3—figure supplement 1A”

“Figure 3—figure supplement 1C” => “Figure 3—figure supplement 1B”

The legends of Figure 3A and Figure 3—figure supplement 1A were modified as follows:

Figure 3A

“Pseudo-color presentation of hair cell loss along the longitudinal axis of the organ of Corti (PND 30, 60, and 120 and noise exposure at PND 60). […] The raw fluorescence image shows the definition of directions (distal and proximal, apex and base) relative to the sensory epithelium.”

Figure 3—figure supplement 1A

“Pseudo-color presentation of the OHC loss frequency along the radial axis. Each row represents a single cochlear sample. […] The radial positions were divided into 15 sections and the scores were averaged within the sections.”

Reviewer #3:[…] Localization and classification of IHCs and OHCs using a machine learning method are nice. But very little details were provided to describe the convolution neural network model and how the training and testing were performed. This needs to be further clarified and fleshed out in the revised manuscript.

The convolutional neural network models were used in the second step of machine learning-based hair cell search. The aim of this step is recovery of false negatives after the first machine learning step based on decision tree ensemble methods (GentleBoost and Random Forest) (the detailed description was inserted in Materials and methods). As suggested, we modified the description about the training and testing of the models in the corresponding section (Table 4; Materials and methods).

It does not look like that the authors have improved the structure of the machine learning model.

As you pointed out, we have not improved the structure of the machine learning model. Instead, we have combined several machine learning algorithms, ensemble learning and convolutional neural network, so as to improve the overall detection efficiency (Materials and methods).

The number of samples used for training and testing the machine learning model is limited and can be improved.

We summarized the sizes of training datasets used for each model as a new table (Table 4). As shown in the table, the datasets include at least several thousand observations, which would be large enough to train the models. As you mentioned, however, these datasets are derived from ten samples, which may not be sufficiently large. Practically, it is not easy to prepare dozens of samples for training, even though our protocol is efficient and automated. We therefore searched the machine learning algorithms that efficiently avoid overfitting with the limited number of samples for training. We added the description about the issue in the corresponding section (Materials and methods). Following your advice, we increased the sample size for test datasets for performance evaluation of the models (Please see Materials and methods).

Since the two-photon fluorescence images have very high contrast, a standard imaging processing method, such as 3D watershed, may just do the job with sufficient accuracy. It would be helpful to include a direct comparison of the machine learning model and the watershed algorithm. The reviewer also felt that the authors may consider downplaying the role of machine learning in the title and throughout the manuscript as traditional methods may achieve similar performance.

As described in our response to the first essential revision above, the approach with 3D watershed could not show enough accuracy of detection on our linearized cochlear images. The main problem was probably the heterogeneity in morphology of hair cells, and the heterogeneity of signal to noise ratio across the image, which make it difficult to set the appropriate thresholds (For details, please see Materials and methods).